# The use of mixed collagen-Matrigel matrices of increasing complexity recapitulates the biphasic role of cell adhesion in cancer cell migration: ECM sensing, remodeling and forces at the leading edge of cancer invasion

**María Anguiano**[1☯], **Xabier Morales**[1☯], **Carlos Castilla**[1], **Alejandro Rodríguez Pena**[1], **Cristina Ederra**[1], **Martín Martínez**[2], **Mikel Ariz**[1], **Maider Esparza**[1], **Hippolyte Amaveda**[3], **Mario Mora**[3], **Nieves Movilla**[3], **José Manuel García Aznar**[3], **Iván Cortés-Domínguez**[1], **Carlos Ortiz-de-Solorzano**[1]*

**1** IDISNA, Ciberonc and Solid Tumours and Biomarkers Program, Center for Applied Medical Research, University of Navarra, Pamplona, Spain, **2** Neuroimaging Laboratory, Division of Neurosciences, Center for Applied Medical Research, University of Navarra, Pamplona, Spain, **3** Department of Mechanical Engineering, Multiscale in Mechanical and Biological Engineering (M2BE), Aragon Institute of Engineering Research (I3A), University of Zaragoza, Zaragoza, Spain

☯ These authors contributed equally to this work.
* codesolorzano@unav.es

**Data Availability Statement:** All raw and processed data used to produce the results shown

## Abstract

The migration of cancer cells is highly regulated by the biomechanical properties of their local microenvironment. Using 3D scaffolds of simple composition, several aspects of cancer cell mechanosensing (signal transduction, EMC remodeling, traction forces) have been separately analyzed in the context of cell migration. However, a combined study of these factors in 3D scaffolds that more closely resemble the complex microenvironment of the cancer ECM is still missing. Here, we present a comprehensive, quantitative analysis of the role of cell-ECM interactions in cancer cell migration within a highly physiological environment consisting of mixed Matrigel-collagen hydrogel scaffolds of increasing complexity that mimic the tumor microenvironment at the leading edge of cancer invasion. We quantitatively show that the presence of Matrigel increases hydrogel stiffness, which promotes β1 integrin expression and metalloproteinase activity in H1299 lung cancer cells. Then, we show that ECM remodeling activity causes matrix alignment and compaction that favors higher tractions exerted by the cells. However, these traction forces do not linearly translate into increased motility due to a biphasic role of cell adhesions in cell migration: at low concentration Matrigel promotes migration-effective tractions exerted through a high number of small sized focal adhesions. However, at high Matrigel concentration, traction forces are exerted through fewer, but larger focal adhesions that favor attachment yielding lower cell motility.

in this paper can be found in Open Access format from the IEEE Data Port repository (DOI: 10.21227/va80-yf12).

**Funding:** This work was supported by the Spanish Ministry of Economy and Competitiveness under grants DPI2015-64221-C2-2 (COdS) and DPI2015-64221-C2-1(JMGA) and fellowships BES-2013-064996 (CC) and BES-2016-076280 (ARP). It was also funded by the Spanish Ministry of Science, Innovation and Universities, under grant RTI2018-094494-B-C22 (COdS). The funders had no role in study design, data collection and analysis, decision to publish, or preparation of the manuscript.

**Competing interests:** The authors have declared that no competing interests exist.

## Introduction

Metastasis is a hallmark of cancer. During the cascade of events that lead to the formation of a metastasis, migrating cancer cells interact with their local microenvironment to sense its biomechanical properties, and adapt in response their migration phenotype. Classically, cancer cell migration dynamics and guidance have been mostly associated with soluble cues such as chemokines [1,2]. Recently, the mechanical factors have been shown to be a key factor in cell migration as well. Consequently, cancer cells adapt their migration phenotype to the properties of the surrounding ECM, or remodel it to exercise a particular form of migration. Indeed, cancer cells remodel their local microenvironment to facilitate the invasion of their surrounding tissue [3]. To this end, cancer cells compact, degrade, cross-link, and align the fibers that form their surrounding local ECM network. Indeed, fiber alignment has been shown to provide directional cues to the cells, directing migration through the anisotropy of the ECM [4], and the alignment of ECM fibers perpendicular to the boundary of a solid tumor has been associated with enhanced invasion in colorectal [5] and breast cancer [6].

Furthermore, deposition and cross-linking of collagens in solid tumors causes increased stiffness of the ECM and is associated with a higher incidence of metastasis [7,8]. Mechanistically, this increased ECM stiffness is linked to both an up-regulation of integrin signaling and increased metalloproteinase activity, which favors the migration of cancer cells, and therefore the spread of the disease. Increased stiffness can also enhance the magnitude of the traction forces exerted by the cells during mesenchymal migration, but the role played by the composition and the structural properties of the matrix in how these forces become effective during cell migration is still unclear.

The role of cell to ECM integrin-dependent interactions, ECM remodeling and traction forces during cancer cell migration has been qualitatively or separately studied, mostly in 2D *in vitro* setups of limited physiological relevance, or in 3D environments devoid of many of the structural proteins and growth factors commonly found in the tumor microenvironment. Here we use a microfluidic 3D platform and mixed collagen-Matrigel hydrogels to quantitatively describe some of the mechanobiological factors that regulate H1299 lung cancer cell migration within a highly physiological environment. The use of increasing concentrations of sarcoma-derived Matrigel, mixed with a fixed concentration of structural collagen, allows us to study the mechanobiology of cancer cell migration in different environments that mimic a normal connective tissue and increasing levels of confinement at the leading edge of tumor invasion [9,10]. In summary, we explain the migratory capacity of these highly metastatic cells [11] in the context of the ECM properties, remodeling and cell-ECM interactions to provide a comprehensive approach to the problem of cancer cell migration.

## Material and methods

### Fabrication of microfluidic devices

Microfluidic devices used to perform H1299 cell migration experiments and ECM remodeling assays were fabricated in polydimethylsiloxane (PDMS) Sylgard 184 by conventional replica-molding process. The master mold was built on 4' silicon wafers by patterning on negative photoresist (SU8-100, MicroChem Co) using standard UV-lithography techniques. The design of the devices is shown in **Fig 1**. The device consists of a main central channel where hydrogels and cells are embedded and two lateral channels that can be used to supply culture medium.

### Collagen I labeling

Rat tail collagen type I (BD Biosciences, San Jose, USA) was labeled with 5-(and-6)-Carboxytetramethylrhodamine, Succinimidyl Ester (5(6)-TAMRA, SE) (Life Technologies, Barcelona,

**Fig 1. Microdevice design.** (A) 2D schematic of the design. (B) PDMS device loaded with blue dye.

Spain) following the method described by Geraldo et al. [12]. Briefly, we injected 1 ml of high concentration collagen (BD Biosciences, San Jose, USA) into a 3 ml dialysis cassette (10,000 MWCO Slide-A-Lyzer TM Dialysis Cassettes) and dialyzed it overnight against a 0.25M sodium bicarbonate buffer (labeling buffer) (Sigma Aldrich, Steinheim, Germany), containing 0.4M sodium chloride at pH 9.5. Then, 100 μl of 10 mg/ml TAMRA solution were mixed with 900 μl of labeling buffer and incubated overnight with rotation with the dialyzed collagen, previously removed from the dialysis cassette. The collagen+TAMRA solution was then dialyzed against the labeling buffer to remove the excess of free dye. The following day, the cassette was dialyzed once more against a solution of 0.2% (v/v) acetic acid (Sigma Aldrich, Steinheim, Germany) in deionized water at pH 4. The concentration of dyed collagen stock was quantified after labeling. The resulting labeled collagen was stored at 4°C protected from light to prevent photobleaching.

## Hydrogel preparation

Hydrogels were prepared using a stock of rat tail collagen type I (BD Biosciences, San Jose, USA) at a final collagen concentration of 2 mg/ml with deionized water, 10x phosphate buffered saline (PBS) (1/10 of the final volume), and NaOH 0.5N, at pH 7. Three types of hydrogels were fabricated; one made of collagen type I and two others made of collagen type I mixed with Matrigel at two increasing concentrations. We refer to them as hydrogels type C (2 mg/ml collagen, no Matrigel), CM (2 mg/ml of collagen, 2 mg/ml of Matrigel), and CM+ (2 mg/ml of collagen, 4 mg/ml of Matrigel), based on the increasing ratio of Matrigel to collagen. TAMRA-labeled hydrogels were prepared using TAMRA-labeled collagen as the collagen.

## Hydrogel microstructural characterization

Image acquisition: TAMRA-labeled hydrogel images were acquired with a 561 nm laser on a Zeiss LSM 800 confocal microscope (Carl Zeiss, Jena, Germany) using a GaAsP PMT detector. 1024x1024x50 voxel image stacks were acquired using an oil-immersion Plan-Apochromatic 63x (1.4 NA) objective lens, for a final resolution of 0.099x0.099x0.42 μm$^3$/voxel. Images were acquired using Zen 2.3 software (Carl Zeiss, Jena, Germany).

Image quantification: Collagen fiber network geometry was analyzed following a method previously described [13,14], which involves a preprocessing filtering step, binarization using local Otsu thresholding, and the use of the FIRE fiber extraction algorithm [15] to obtain a 3D reconstruction of the collagen networks that allows the quantification of the following morphological parameters: average fiber length, persistence length, and network pore size.

Nine representative images, corresponding to three image stacks acquired from three independent replicas of each of the hydrogel types (C, CM, CM+), were analyzed as described above. The comparison of the parameter values obtained for the three types of hydrogels was statistically analyzed using a one-way ANOVA followed by Bonferroni corrected post-hoc test.

## Hydrogel rheological characterization

The mechanical properties of the hydrogels were characterized by rheological assays using a stress-controlled 35 mm diameter rotational rheometer of plate-cone geometry HAAKE Rheostress 1 (Thermo Fisher Scientific, Waltham, MA, USA), following the protocol described by Valero *et al.* [16]. Briefly, 300 µl of hydrogel prepolymer were prepared and pipetted on the lower plate of the rheometer. Then, the upper plate descended until the gap between both plates was the required by the sensor specifications (0.051 mm) The hydrogel was surrounded by low viscosity oil (0.1 Pa.s) to avoid hydrogel dehydration. Then, the hydrogel prepolymer was allowed to polymerize and stabilize at 37˚C for several hours while applying cyclic strain of amplitude 0.5% and frequency 0.1 Hz before oscillatory stress sweep assay while instantly recording the shear modulus of the sample. After hydrogel stabilization, a torque of 5 µNm (at a frequency $\gamma = 0.1$ Hz) was applied to the sample and stepped up until hydrogel failure.

Three hydrogels of each type (C, CM, CM+) were independently characterized. The storage G' and loss moduli G" of each hydrogel were measured over time. The measurements obtained were averaged per hydrogel type. The obtained G' and G" values were compared between hydrogels using a Kruskal-Wallis test.

## Cell culture, maintenance, and transfection

The H1299 non-small lung cancer cell line was purchased from the American Type Culture Collection (ATCC, LGC-Promochem SL, Barcelona, Spain). Cells were cultured in RPMI medium (GIBCO, Barcelona, Spain) supplemented with 10% fetal bovine serum (FetalClone III, Thermo Fisher Scientific, Madrid, Spain) and 1% of a combination of penicillin and streptomycin (100 units/ml).

When required, H1299 cells were transfected with the pLE-GFP-C1 plasmid (Mountain View, CA, USA) or the mCherry-vinculin-23 plasmid (gently donated by Michael Davidson, Addgene plasmid#55159) using lipofectamine reagent (Life Technologies, Barcelona, Spain) according to the manufacturers' protocols. After transfection, positive cells were selected in the presence of G418 antibiotic (Sigma Aldrich, Steinheim, Germany) at a 1 mg/ml concentration.

## Cell migration assays in microfluidic devices

To perform cell migration experiments, cells were centrifuged and resuspended in the hydrogels at a concentration of 1000 cells/µL. Before hydrogel filling, the microdevices were coated using Poly-D-lysine and sterilized under UV light during 20 minutes. Then, 10 µl of the cell-containing hydrogel prepolymer was pipetted in one of the inlets of the hydrogel channel and gently pushed until the central device chamber was completely filled. Then, the device was incubated at 37˚C and 5% $CO_2$ for 40 minutes to allow hydrogel polymerization. Afterward, the lateral channels were filled with 20% FBS medium, and the video sequence acquisition started.

For the MMP-blocking experiments, cells were pre-incubated with GM6001 metalloproteinase inhibitor (Millipore, Billerica, MA, USA) at a concentration of 25µM for 2 hours in serum-free cell culture medium before embedding the cells in the hydrogels. After that the microdevices lateral channels were filled using serum-free medium or 20% FBS medium with GM6001 25µM to maintain MMP blockade during image acquisition.

Microdevices were imaged in fluorescence and phase contrast microscopy using a Zeiss CellObserver SD spinning disc confocal microscope (Carl Zeiss, Jena, Germany) equipped with a dry Plan-Apochromatic 5x (0.12 NA) objective lens. 2D time-lapse videos of migrating cells fully embedded in the hydrogels were obtained by capturing images every 15 minutes for

12 hours. To quantify cell motility, 2D tracks of migrating cells were calculated, using the acquired fluorescence time-lapse videos as described by Anguiano *et al.* [11]. First, a contrast limited adaptive histogram equalization filter was applied to enhance image contrast [17]. Then, the cells were segmented using the graph cut approach by Maska *et al.* [18], which segments objects based on their mean intensity. The segmented cells were associated between successive frames using a constrained nearest-neighbor approach, implemented in the CellTracker system [19].

We calculated the mean accumulated distance (MAD) and the average speed of at least 200 cells in 4 independent videos of each hydrogel type and treatment (Control or GM6001). The comparison between the MAD migration results obtained with and without GM6001 treatment, for a given hydrogel type, was performed using a U-Mann-Whitney test.

## Invasion assays in Boyden chambers

Cells were embedded in the hydrogel at a concentration of 1000 cells/μl, and placed in the upper compartment of an 8 μm pore sized Boyden chamber insert (Corning, New York, USA). Afterward, hydrogel was incubated at 37˚C for one hour to polymerize. Cell invasion was stimulated by filling the lower compartment of the chamber with 20% serum supplemented cell culture medium. After 48 hours at 37˚C in the incubator, cells were fixed in 4% formaldehyde for 15 minutes and the upper side of the insert was thoroughly wiped off using cotton swabs. The cells in the lower part of the membrane were stained with 0.5% Crystal Violet dye for 20 minutes at room temperature (Sigma Aldrich, Steinheim, Germany).

To perform invasion assays under matrix metalloproteinase (MMP) blocking, cells were pre-treated with the MMP inhibitor GM6001 (Millipore, Billerica, MA, USA) for 2 hours at a concentration of 25μM in serum-free medium before being embedded in the hydrogels. After that, MMP blockade was maintained during cell invasion by placing serum free medium containing GM6001 in the upper compartment of the Boyden chamber.

After staining, 2D images were acquired using a Leica DMIL led inverted microscope (Leica Microsystems, Wetzlar, Germany), with a HI Plan 10x (0.22 NA) objective lens and equipped with a Leica EC3 digital camera. Five (5) hydrogels of each type and treatment were used, in which four representative fields were acquired per Boyden membrane (n = 20 images). Invasion assays were quantified by counting and normalizing the number of stained cells per experiment. The obtained values were compared using a one-way ANOVA followed by Bonferroni post-hoc test.

## Fiber alignment and ECM compaction assays

To determine the ability of migrating H1299 cells to align and compact their surrounding collagen fibers, microfluidic devices were filled with TAMRA-labeled hydrogels seeded with CellBrite-labeled cells at a density of 1000 cell/ul. After polymerization, one lateral channel was filled with serum free culture medium and the other one with RPMI medium supplemented with 20% FBS. Hydrogel remodeling was imaged using an inverted confocal microscope ZEISS LSM 800 (Carl Zeiss, Jena, Germany) equipped with a 40x immersion oil objective lens, 24 hours after loading the microfluidic devices. Z-stacks of 512x512 pixels image size of the cells and their surrounding collagen fibers were acquired at a resolution of 0.312x0.312x0.49 μm.

Fiber anisotropy was quantified in 512x512 pixel images of the hydrogels. The anisotropy index, an indicator of fiber alignment, was measured in a control alignment ROI of 100x100 pixels located in a non-cellular region of the image, and in a 100x100 pixel ROIs in-between cells or in close vicinity of cells. To obtain the anisotropy index, the Fourier Transform Method

described by Sander *et al*. [20] was used: an isotropic network made of randomly oriented fibers has an anisotropy coefficient value of zero, while a completely aligned network has the maximum value of anisotropy which is one.

For each hydrogel type (C, CM, CM+), three independent hydrogels were analyzed. To this end, seven representative images were acquired per hydrogel, from which one measurement of the control and alignment anisotropy indices was computed (n = 21 images). The comparison of the anisotropy index between hydrogels was analyzed using a one-way ANOVA followed by Bonferroni corrected post-hoc test.

ECM densification was assessed by comparing the fiber to background ratio in binarized ROIs of the images. The percentage of pixels occupied by collagen fibrils in the areas of interest was used as the indicator of collagen fibril density. Images were binarized using the Moments threshold algorithm in Fiji. Fiber to background ratio was calculated in a control density ROI of 100x100 pixels located in a non-cellular region of the image, in a 100x100 pixel ROI of fiber alignment in-between cells or in close vicinity of cells, and in "Doughnut" shape ROIs of 7 μm thickness in the surrounding area of each cell. "Doughnut ROIs" were selected using the Fiji command Make Band Selection, by specifying the desired thickness of the ROI at the boundary of cells.

For each hydrogel type (C, CM, CM+), three independent hydrogels were analyzed. To this end, seven representative images were acquired per hydrogel, from which one measurement of the control, alignment and surrounding collagen density measurements was computed (n = 21 images). The comparison of the collagen densification values between hydrogels was performed using a one-way ANOVA followed by Bonferroni corrected post-hoc test.

## DQ-collagen I degradation assays

To study hydrogel remodeling, hydrogels were prepared with a cell density of 1000 cells/μl and DQ-collagen type I (Life Technologies, Invitrogen, Barcelona, Spain) at 25 μg/ml following the protocol described by Pelaez *et al*. [21]. Cells were labeled with CellBrite before seeded in the hydrogel. Once prepared, the hydrogel was delivered into the central channel of a microfluidic device and was incubated for 40 minutes at 37˚C to allow hydrogel polymerization. Then, the microdevice's lateral channels were filled with cell culture medium; one channel was filled with serum-free culture medium and the other one with RPMI medium supplemented with 20% FBS. Hydrogel remodeling was imaged using an inverted confocal microscope ZEISS LSM 800 (Carl Zeiss, Jena, Germany), 24 hours after loading the device. A 40x immersion oil objective lens was used to acquire Z-stacks of the cells and their surrounding collagen fibers and DQ-collagen degradation signal in two different emission channels.

Proteolytic degradation of hydrogels during cell migration was quantified as the volume of the dye-quenched protein substrate (DQ-collagen type I). To calculate this volume, image stacks were first binarized using the Moments threshold algorithm that implements Tsai's moment preserving approach [22]. This method selects a threshold such that the binary image has the same first three moments as the grey level image. Following binarization, a morphological closing operator was applied to the images, and they were then filtered with a median filter to remove unwanted noise.

Three independent hydrogels were analyzed per type of hydrogel and treatment. DQ-collagen degradation was then measured in ten representative images of each hydrogel (n = 30 images). DQ-volume results with and without GM6001 treatment were compared using a one-way ANOVA followed by Bonferroni corrected post-hoc test.

## Vinculin labeling

To quantify cell adhesions onto 2D hydrogels, H1299 cells were seeded on top of the hydrogels in 8-well slides (Labteck, Nunc, Denmark). After hydrogel polymerization, RPMI medium (GIBCO, Barcelona, Spain) supplemented with 20% FBS was added to the wells and they were allowed to adhere for 6 hours. Afterward, samples were rinsed off with PBS and fixed in 4% PFA at 37˚C for 15 minutes. Cells were permeabilized with 0.005% Triton X-100 in PBS, washed with PBS, and incubated overnight at 4˚C with the primary antibody Anti-Vinculin (clone H-Vin 1, Sigma-Aldrich, Steinheim, Germany) at a dilution 1:200. The following day, samples were washed with PBS and subsequently incubated for one hour at room temperature with a secondary Donkey Anti-Mouse antibody conjugated with AlexaFluor 488 (1:400, Thermo Fisher, Waltham, MA, USA). Finally, 2D images were acquired with an oil immersion 63x Plan-Apochromat objective (1.4 NA) on a Zeiss LSM 800 laser-scanning confocal microscope (Carl Zeiss, Jena, Germany). Images were acquired using Zen 2.3 software (Carl Zeiss, Jena, Germany).

The location and distribution of focal adhesions within 3D hydrogels was assessed using the H1299 cell line stably transfected with a mCherry-vinculin plasmid. Briefly, cells were embedded in each hydrogel (C, CM, CM+) at a concentration of 1000 cells/µl, and placed onto 8-well Labtek chambers (Labtek, Nunc, Denmark). After polymerization, cell migration was stimulated by placing FBS-containing cell culture medium on top of the hydrogel. Then hydrogels were then incubated overnight at 37˚C. The following day, the clusters of mCherry-vinculin at cell protrusions were acquired with an oil immersion 63x Plan-Apochromat objective (1.4 NA) equipped on a Zeiss LSM 800 laser-scanning confocal microscope (Carl Zeiss, Jena, Germany). Images were acquired using Zen 2.3 software (Carl Zeiss, Jena, Germany) and 3D renderings were performed using Amira 5.2v (Thermo Fisher Scientific, Waltham, MA, USA).

## Quantification of vinculin-stained focal adhesions in H1299 cells

2D images of vinculin stained cells were processed and quantified using the Fiji software [23], following the method described by Horzum *et al.* [24]. Prior to image processing, coarse binary masks of the cells were obtained to roughly define the areas where to search for focal adhesions (FAs) in the images. The raw images were processed as follows: first, a median filter was applied and the background was subtracted using the subtract background tool with the sliding paraboloid option. Then, contrast was enhanced by running the CLAHE (Contrast Limited Adaptive Histogram Equalization) plugin [17] and the background was further reduced by applying the mathematical exponential function (EXP). Subsequently, the brightness and contrast of the images were automatically adjusted and an edge-enhancing Laplacian of Gaussian filter [25] was applied. Images were automatically thresholded within the cell mask to detect FAs, followed by a morphological closing to merge loosely separated FA fragments. Finally, the analyze particles command was used to identify and automatically measure the size of the detected focal adhesions. All these steps were included in an in-house developed plugin that automates the quantification. The parameters used in the quantification are listed in **Table 1**.

Three independent hydrogels of each type of hydrogel were fabricated, and a total of 21 representative images containing at least one cell were acquired and analyzed as described above. The comparison of the mean area of focal adhesions between 2D substrates was performed using a Krustal-Walllis test. FA detection and quantification in H1299-mCherry-vinculin cells embedded in the 3D hydrogels was carried out following a home-made plugin developed for Fiji. Briefly, the 3D image stacks were denoised by applying a 3D median filter (1-voxel radius).

**Table 1. Parameters used for 2D FA quantification.**

| Step | Parameter | Value |
|---|---|---|
| Median filter | Radius | 2 |
| CLAHE | Blocksize, histogram, slope | 19, 256, 6 |
| Enhance Contrast | Saturation | 0.35 Automatic |
| LoG 3D | σx, σy | 5, 5 pixels |
| Automatic threshold | Default | Default |
| Closing | Default | Default |
| Analyze particles. | Size, circularity | 50-Infinity, 0–1.0 |

Then background subtraction was performed by applying a rolling ball algorithm with a 10-pixel radius to the entire stack. Afterward, the contrast of the FAs was enhanced by applying a non-linear contrast adjustment with a gamma value of 2.1. The FAs in the resulting stack were segmented using a fixed manual threshold that selects the hyperintense FAs, followed by a grayscale watershed segmentation of the Euclidian Distance Map (EDM) with adjustable sensitivity to split clustered FA.

The resulting 3D segmentation mask was analyzed using MorphoLibJ [26], which implements a 3D component analysis and quantifies the number of FAs and their individual volumes. Please note that the vinculin expression associated with the Golgi apparatus and reticulum was subtracted from the segmentation mask before FA quantification, by applying a fixed size 5-micron band around each nucleus.

Three independent hydrogels of each type of hydrogel were analyzed, and a total of 25 representative images containing one cell were acquired and analyzed as described above. The comparison of the average number of FAs per cell, between cells embedded in the different 3D substrates was performed using a one-way ANOVA followed by Bonferroni corrected post-hoc test. The comparison of the mean area of focal adhesions between cells embedded in the different 3D hydrogels was performed using a Krustal-Wallis test.

## In vivo cell labeling

H1299 wild-type cells were labeled using the cytoplasmic membrane dye CellBrite™ Red 644/665 nm (Biotium) following manufacturer's staining protocol. Cells were suspended in complete medium at a density of $10^6$ cells/ml. Then, 1 µl of CellBrite was added to the cells in suspension and thoroughly mixed by flicking the tube. After 20 minutes of incubation at 37˚C, the labeled suspension was centrifuged for 5 minutes at 1500 rpm at 37˚C. The supernatant was discarded, and cells were gently resuspended in 1 ml pre-warmed complete medium. The washing procedure involving centrifugation and resuspension in complete medium was repeated two more times before using the cells.

## Flow cytometry assays

To determine H1299 expression of β1 and β3 integrins at the membrane in hydrogels C, CM, and CM+, cells were embedded in the hydrogels at a concentration of 1000 cells/µl. The hydrogel prepolymer were loaded to each well of an 8-well Labtek chambers (Lab-Tek, Nunc, Roskilde, Denmark). Then, the hydrogel was allowed to polymerize for 40 minutes at 37˚C and afterward, cell migration was stimulated by adding 20% serum supplemented cell culture medium and incubated overnight at 37˚C. The following day, cells were recovered from the hydrogels using cell recovery solution (Corning, Bedford, MA, USA) by following manufacturer's instructions.

To quantify β1 and β3 integrin surface expression, cells were placed in a 96-well round-bottom microwell (Sigma Aldrich, Steinheim, Germany) and were incubated with primary antibody at 4˚C for 20 minutes. β1 expression was assessed using Anti-Integrin β1 antibody (Abcam, AB7168), whereas β3 expression was analyzed using Anti-Integrin β3 antibody (Millipore, MAB2023Z, Billerica, MA, USA). Both primary antibodies were diluted at 1 μg/ml in PBS prior to incubation. Isotype control was performed using Anti-Mouse IgG1 antibody (1:100, Biolegend, San Diego, USA).

After primary antibody incubation, cells were washed with cold PBS and incubated with the secondary antibody AlexaFluor 488 (1:400, Invitrogen, Barcelona, Spain) for 10 minutes at 4˚C. Subsequently, the excess of secondary antibody was washed with PBS and cells were resuspended in 500 μl of PBS at 4˚C containing 5mM EDTA (Lonza, Barcelona, Spain). Fluorescence measurements and data analysis were performed using BD FACSCalibur (BD Biosciences, San Jose, USA) cytometer and FlowJo software, respectively. Three independent hydrogels of each type (C, CM, CM+) were fabricated and at least 50.000 cells were analyzed per condition.

## Cell adhesion assays to ECM proteins

H1299 cell adhesion assays were performed in 96-well flat-bottom cell culture plates (Sigma Aldrich, Steinheim, Germany). Wells were coated with type I collagen at 50 μg/ml, fibronectin at 50 μg/ml, collagen IV at 50 μg/ml and 3% BSA (w/v), used as negative adhesion control (Sigma Aldrich, Steinheim, Germany). After adding the coating solutions plate was incubated for 2 hours at 37˚C to polymerize. Then, wells were rinsed with PBS and subsequently blocked with 1% (w/v) BSA for 1 hour at room temperature. Afterward, excess BSA was removed from the wells and 20.000 cells were added to each coated well in adhesion cell culture medium, consisting of RPMI medium supplemented with 0.5% (w/v) BSA and HEPES 20mM. Cells were allowed to adhere to the wells for 30 minutes at 37˚C and then the wells were washed with PBS to remove the cells that had not adhered to the bottom of the wells. Cells were fixed with 4% paraformaldehyde (PFA) (AppliChem, Darmstadt, Germany) for 20 minutes at room temperature and stained with 0.5% (w/v) Crystal Violet dye (Sigma Aldrich, Steinheim, Germany) for 20 minutes at room temperature.

After Crystal Violet staining, samples were rinsed off with water to remove excess of dye and dried at room temperature. Cells were lysed using 2% SDS (v/v) in water for 30 minutes under agitation. Finally, to determine the percentage of adhesion to the different ECM proteins, the absorbance at 490 nm was measured using a 96-well plate reader (TecanSunrise, Tecan, Männedorf, Switzerland). Results were corrected against the absorbance obtained in the BSA coated wells. To determine 100% adhesion for each ECM protein, absorbance was measured at 490 nm in their control wells that contain the original 20.000 cells. Four wells were analyzed per ECM protein and experiment. The experiment was performed three times (n = 12 wells).

## Traction force microscopy assays

C, CM, and CM+ hydrogels were seeded with cells at a density of 15.000 cells/well and plated onto 8-well Labtek chambers (Labtek, Nunc, Roskilde, Denmark). Hydrogels were incubated overnight with FBS-containing cell culture medium to allow cell adherence. Each well was loaded with 300 μl of hydrogel, and FBS-containing medium was added after polymerization. Then, the embedded cells were treated with a drug mix that contained 1mM Nocodazole (Sigma Aldrich, Steinheim, Germany), an antimitotic drug that disrupts microtubules and inhibits microtubule dynamics, causing cell protrusions to retract, and 50μM Blebbistatin

(Sigma Aldrich, Steinheim, Germany), a non-muscle myosin-II inhibitor that impairs actin cytoskeletal tension, causing cell relaxation.

Starting at the time of treatment, cell-containing hydrogel volumes were imaged every 20 minutes for 2 hours, until reaching the force-free relaxed state of the hydrogel, using a Zeiss LSM 880 AxioObserver laser scanning microscope (Zeiss, Germany) and a 40x Plan-Apochromat 0.95 NA objective. Label-free confocal reflection images of the hydrogel collagen fibers were acquired using a 633 nm laser, and the H1299 GFP cells were imaged using the 488 nm laser. The acquired Z-stacks had a FOV of 212x212x~50 microns (depending on its size and protrusions length), with a voxel size of 0.42x0.42x0.50 $\mu m^3$. The microscope was equipped with a motorized stage placed on an anti-vibration table, located in a room kept at a constant temperature of 21˚C. Cells were kept at 37˚C and 5% CO2 in the microscope incubation chamber during the entire length of the experiments.

For each hydrogel type and cell analyzed, the unconstrained force-induced deformation was measured using the SAENO software [27]. SAENO takes as input the images of the pre-treatment (time 0) and post-treatment, relaxed hydrogels. We measured traction forces exerted by H1299 GFP cells embedded in C, CM, and CM+ hydrogels, with a shear elastic modulus ranging from 11 Pa in C hydrogels to 32 Pa in CM+ hydrogels. The parameters used to calculate cell contractility are listed in **S1 Table**, where $K_0$ (bulk modulus of the material), $D_0$ (buckling coefficient), $L_S$ (onset of strain stiffening) and $D_S$ (strain stiffening) are the main parameters that describe the properties of the collagen network. At least 5 cells (from independent hydrogel samples) were analyzed per hydrogel type. The comparison of contractility values between cells embedded in the three different hydrogels was perfomed using a one-way ANOVA followed by Bonferroni corrected post-hoc tests.

## Actin cytoskeleton labeling

H1299 wild-type cells were transiently transfected to express the F-actin protein conjugated with the green fluorescence protein (GFP). The transfection was done with the expression vector LifeAct-GFP (Ibidi, Martinsried, Germany) using lipofectamine reagent (Life Technologies, Barcelona, Spain) following the manufacturer instructions. LifeAct-GFP readout started after 24 to 48 hours from transfection.

## High-resolution analysis of filopodial protrusions in migrating H1299 cells

LifeAct transfected H1299 cells were mixed up with any of the three different hydrogel types (C, CM, and CM+) at a concentration of 1000 cells/$\mu$l. Then, 200 $\mu$l of the hydrogel-cells mix was gently inserted into an 8 well chamber and incubated for 45 minutes at 37˚C and 5% CO$_2$. Once the hydrogel was polymerized, cells were treated with 1mM Nocodazole to arrest the tubulin cytoskeleton. Then, the Nocodazole solution was removed and 20% serum was placed over the hydrogel to promote microtubule polymerization and cell protrusions formation. Samples were placed in the incubation chamber of the Zeiss LSM 880 laser-scanning confocal microscope (Carl Zeiss, Jena, Germany) using an oil immersion 63x Plan-Apochromat (1.40 NA) objective lens, a GaAsP and two PMT detectors. 3D time-lapse videos of migrating cells were acquired every 2 minutes for 1 hour resulting in 30-frames videos. Cells were excited with the 488 nm laser line, and fluorescence images were acquired with a 63x objective lens. The cells were captured with a fine Z-axis spacing of 0.5 $\mu$m being the total Z-stack size adjusted to each cell, depending on its size and protrusions length (~70 $\mu$m).

Cell protrusions were confirmed of filopodial nature by co-staining with phalloidin, a classical F-actin marker, and fascin, an actin-adaptor protein specifically localized at filopodia. To that aim, cells were embedded within the three hydrogels types and treated as described above

to stimulate cell protrusions formation. Afterward, cells were fixed in 4% PFA at 37°C for 30 minutes and permeabilized with 0.005% Triton X-100 for 15 minutes at room temperature. Non-specific interactions were blocked with 3% BSA and subsequently were incubated overnight at 4°C with the primary antibody Anti-Fascin (clone 55k2, Millipore, Billerica, MA, USA) at a dilution 1:200. The following day, samples were rinsed with PBS and incubated for two hours at room temperature with a secondary Donkey Anti-Mouse antibody conjugated with AlexaFluor 488 (1:400, Thermo Fisher, Waltham, MA, USA) and rhodamine-labeled phalloidin (1:75, Sigma Aldrich, Steinheim, Germany). Finally, Z-stack images were acquired with an oil immersion 63x Plan-Apochromat objective (1.4 NA) on a Zeiss LSM 800 laser-scanning confocal microscope (Carl Zeiss, Jena, Germany). Images were acquired using Zen 2.3 software (Carl Zeiss, Jena, Germany).

## Results

### Matrigel increases the pore size and heterogeneity of the underlying mesh

The quantification of the morphology of the TAMRA-stained collagen mesh in all three hydrogel types (Fig 2) is shown in Table 2. These values show larger pore sizes in hydrogels with the highest Matrigel concentration (CM+) compared to collagen only hydrogels (C) or hydrogels with an intermediate Matrigel concentration (CM) ($p < 0.01$). Fiber length was also longer in CM+ than in C or CM hydrogels ($p < 0.05$, $p < 0.01$, respectively). This seems to indicate that Matrigel favors the formation of collagen fiber bundles during hydrogel polymerization, which in turn causes fewer, but larger pores and a heterogeneous collagen mesh.

### Matrigel addition produces stiffer hydrogels and stronger strain-hardening behavior in mixed collagen-Matrigel hydrogels

The oscillatory stress sweep assays (Fig 3) revealed a strain-stiffening phenomenon in all three hydrogels. This effect is however clearly stronger in hydrogels CM+ compared to C or CM hydrogels. The G' and G'' values measured right after hydrogel polymerization and the maximum values reached at the end of the stress sweep assay are shown in Table 3. The G' values

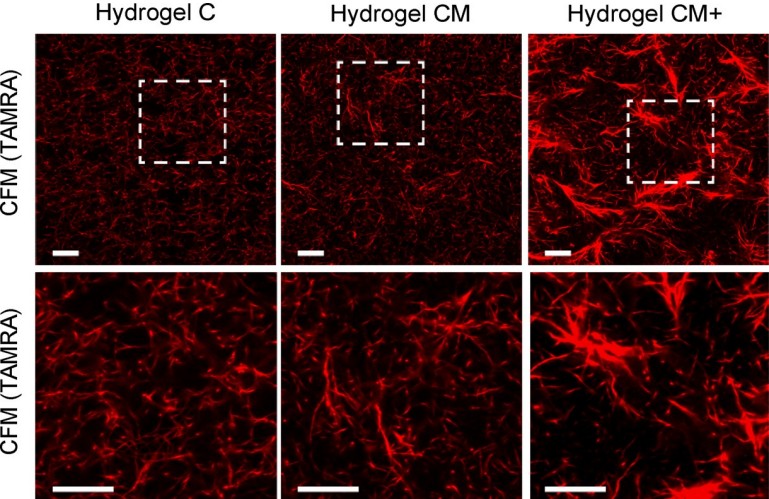

**Fig 2. Hydrogel morphology.** Representative confocal fluorescence microscopy images obtained from C, CM, and CM+ hydrogels labelled with TAMRA. White insets show representative magnified areas of the collagen mesh. (Scale bar = 10 μm).

**Table 2. Morphological characterization of hydrogels.**

| Hydrogel | Fiber length | Fiber persistence | Pore size (μm) |
|---|---|---|---|
| C | 4.35 (0.03) | 2.24 (0.02) | 1.91 (0.08) |
| CM | 4.06 (0.09) | 2.21 (0.07) | 2.50 (0.05) |
| CM+ | 4.99 (0.25) | 2.63 (0.14) | 3.04 (0.14) |

Average and standard error (SEM) of the morphological measurements obtained from the confocal images of TAMRA-labeled hydrogels C, CM, and CM+. (Number of samples n = 9)

measured immediately after polymerization show that CM+ hydrogels are stiffer than CM and C hydrogels (p<0.05). In summary, Matrigel contributes to hydrogel stiffening and favors a strong non-linear strain-hardening phenomenon in our collagen-based hydrogels.

## Metalloproteinase-blocking reduces cell migration in Matrigel-containing hydrogels

To measure the extent of H1299 lung cancer cell migration in environments of different composition, cell migration experiments were performed in microfluidic devices containing our hydrogels C, CM, and CM+, free or under the effect of metalloproteinase-blockade. The results are shown in **Fig 4** and **Table 4**. As previously reported by us [11], our experiments confirmed that cancer cells with normal metalloproteinase activity, embedded in CM hydrogels, migrate longer distances than in hydrogels of C or CM+ type (p<0.05). We now show also reduction in cell migration in Matrigel containing hydrogels CM (p<0.05) and CM+ (p<0.05) when

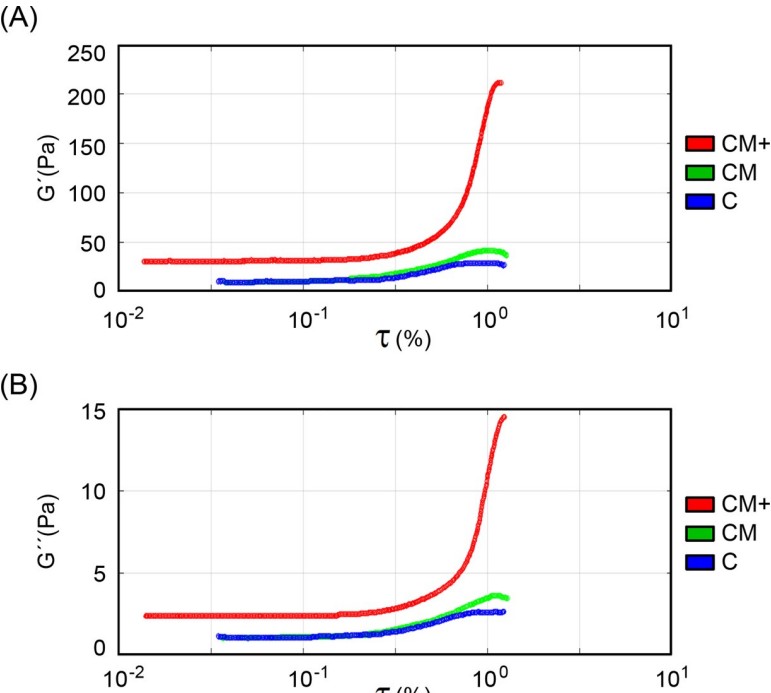

**Fig 3. Hydrogel rheological characterization.** Rheological assays performed on C, CM, and CM+ hydrogels. (A) Shear modulus during stress sweep rheological assays. (B) Loss modulus during the stress sweep rheological assays. (Number of experiments, n = 3).

**Table 3. Rheological characterization of hydrogels.**

| Hydrogel | G' (Pa) | G" (Pa) | G' Max (Pa) | G" Max (Pa) |
|---|---|---|---|---|
| C | 11.64 (0.15) | 1.17 (0.04) | 30.22 (2.19) | 2.75 (0.07) |
| CM | 10.95 (0.27) | 1.06 (0.03) | 42.84 (1.63) | 3.65 (0.18) |
| CM+ | 31.77 (1.29) | 2.52 (0.15) | 213. 53 (4.22) | 15.91 (0.53) |

Average and standard error (SEM) of the rheological values measured after polymerization (G', G") and maximum values obtained (G' Max and G" Max) during the stress sweep assays. All values are given in Pa. (Number of samples used to calculate the moduli, n = 3).

cells are treated with the GM6001 MMP inhibitor, pointing at a relevant role of proteolytic matrix degradation during H1299 cancer cell migration in Matrigel containing hydrogels.

## Metalloproteinase-blocking reduces invasion in Matrigel-containing hydrogels

We next looked at the role of metalloproteinases in H1299 lung cancer cell invasion using Boyden chambers filled with the three hydrogels under study. The invasion results using cells free or under the effect of MMP inhibitor GM6001 are presented in **Fig 5** and **Table 5**. These findings confirm the results of our cell migration experiments. Indeed, reduced invasion, caused by MMP-blockade, was observed both in hydrogels CM and CM+ ($p < 0.001$), pointing at a relevant role of metalloproteinases in cancer cell migration within hydrogels of complex composition.

## Matrigel induces strong hydrogel remodeling, measured as the alignment and densification of the collagen mesh surrounding H1299 cancer cells

We looked quantitatively at the amount of hydrogel remodeling caused by migrating H1299 cells, as a function of hydrogel composition. A schematic representation and one example of the images and regions used to calculate fiber alignment and collagen densification are shown in **Fig 6A** and **Fig 6D**, respectively. The anisotropy index α, which quantifies fiber alignment, was first measured in control ROIs, located at non-cell containing regions of the images. Then it was also measured in areas of fiber alignment located between cells (alignment ROIs). Both

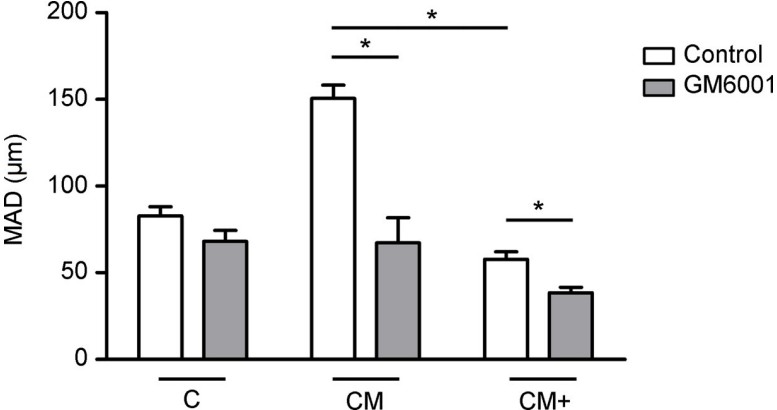

**Fig 4. Cell migration assays within microfluidic devices.** Quantification of cell migration (MAD) in hydrogels C, CM, and CM+ in not treated experiments (Control) and with MMP-blocking treatment (GM6001). *Indicates statistically significant difference between groups ($p < 0.05$). (Number of videos for each type of hydrogel, n = 4).

**Table 4. H1299 migration experiments inside the microfluidic devices.**

| Hydrogel | C | CM | CM+ |
|---|---|---|---|
| | CONTROL | | |
| MAD (μm) | 82.73 (5.40) | 150.52 (7.74) | 57.80 (4.35) |
| Speed (μm/h) | 6.89 (0.45) | 12.54 (0.64) | 4.82 (0.18) |
| | GM6001 | | |
| MAD (μm) | 68.15 (6.23) | 67.25 (14.55) | 38.49 (3.15) |
| Speed (μm/h) | 5.68 (0.52) | 5.60 (1.21) | 3.21 (0.26) |

H1299 migration experiments inside our microfluidic devices. Mean and standard error (SEM) of accumulated distance (MAD) in microns, and speed of migration (μm/h) within hydrogels C, CM, and CM+. Experiments were performed under 20% FBS stimulation with (GM6001) our without (Control) MMP-blocking treatment. (Number of videos for condition and type of hydrogel, n = 4).

the control and alignment ROI anisotropy values are shown in **Table 6** and in **Fig 6B**. The control anisotropy index was similar in all three hydrogels, being always less than 0.2. The alignment anisotropy was higher than the control anisotropy in all hydrogels (p<0.001), indicating that cells, to migrate, align the collagen fibers in their surroundings. However, the level of alignment anisotropy was higher in Matrigel-containing hydrogels CM and CM+ compared to collagen only, C hydrogels (p<0.001).

Matrix densification results are presented in **Table 7** and **Fig 6C**. The density of collagen fibers in control areas was similar in all three hydrogels, being always less than 0.25. The density measured in the surrounding area of cells (Doughnut ROIs) was higher than the density

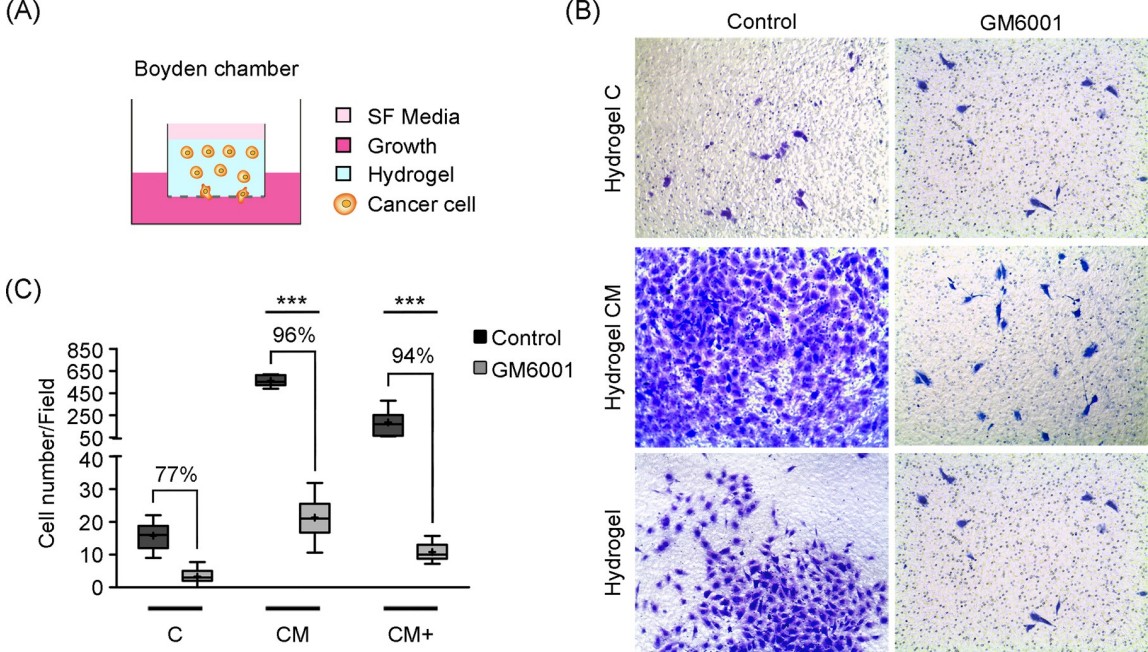

**Fig 5. Cell invasion assays within Boyden chambers.** Boyden invasion assays in control experiments and with GM6001 MMP-blocking treatment. (A) Diagram of the Boyden invasion assay. Cells migrate through the hydrogel and pores of the transwell membrane in response to 20% FBS cell culture medium. (B) Representative images used to quantify cell invasion. (C) Box-plot of the number of invading cells in control and GM6001 invasion experiments. *** Indicates highly statistically significant difference between groups (p<0.001). (Number of images, n = 20).

**Table 5. Hydrogels invasion assays within Boyden chambers.**

| Hydrogel Type | Control | GM6001 |
|---|---|---|
| C | 15.75 (1.48) | 3.6 (0.43) |
| CM | 560.63 (17.31) | 22.25 (1.22) |
| CM+ | 186.43 (43.72) | 11.1 (0.52) |

Number of invading cells and standard error (SEM) in H1299 invasion experiments in Boyden chambers. Experiments were done under 20% FBS stimulation with (GM6001 treatment) or without (control) MMP-blocking treatment. (Number of images, n = 20).

measured in control areas in all hydrogels (p<0.001), demonstrating that cells compact the collagen fibers in the three types of hydrogels. Furthermore, in hydrogels CM and CM+, the compaction of collagen around cells was higher than in the alignment areas (p<0.001), being also higher than the values measured around cells in collagen only, C hydrogels (p<0.001). These values show that ECM remodeling, measured by the compaction and alignment of fibers exists in all hydrogels, but is more prevalent in Matrigel containing than in collagen only hydrogels.

## H1299 cells cause strong matrix degradation in Matrigel-containing hydrogels

Matrix degradation caused by migrating H1299 cancer cells was measured using DQ-collagen I assays. Representative 2D and 3D images of the confocal Z-stacks used to measure collagen degradation are shown in **Fig 7B–7E**. The quantification of collagen degradation ($\mu m^3$/cell) in all three hydrogel types, with and without the effect of Metalloproteinase inhibitor GM6001 is shown in **Fig 7A**. Our results show that the level of collagen degradation by proteolytic activity was larger in CM+ hydrogels compared to CM and C hydrogels (p<0.001). CM hydrogels also showed stronger degradation than C hydrogels (p<0.001), highlighting that H1299 cancer cell migration in Matrigel containing hydrogels requires higher levels of MMP activity. Consistently, treatment of cells with GM6001 reduced the volume of DQ-collagen degradation in hydrogels CM and CM+ (p<0.001) but not in hydrogel C (p>0.05), indicating that cells in collagen-only C scaffolds migrate using a low MMP-dependent mechanism. In summary, in pure collagen hydrogels, H1299 cells barely degrade the surrounding matrix, whereas, the presence of Matrigel in these hydrogels induces larger proteolytic activity and generation of microtracks that facilitate cell mobility.

## H1299 cells display a higher number of larger focal adhesions in Matrigel-containing hydrogels compared to collagen only hydrogels

Focal adhesions (FAs) were first quantified in H1299 cells adhered to hydrogels C, CM, and CM+. Representative images of the vinculin immunostaining of H1299 cells are shown in **Fig 8A**. The number and area of the FAs is shown in **Fig 8B and 8C**. As shown, the number of FAs was higher for cells on hydrogel CM, that contains an intermediate concentration of Matrigel, compared to hydrogels C and CM+ (p<0.001, p<0.01 respectively). However, the average area of the focal adhesions increased with increasing Matrigel content, being the highest in CM + hydrogels (p<0.001). Therefore, both the number and size of FAs increases due to the presence of Matrigel, although a different behavior is seen regarding the number and size of the adhesions as Matrigel concentration increases. An intermediate concentration seems to favor a higher number of smaller FAs, while at high concentrations favors fewer but larger, more

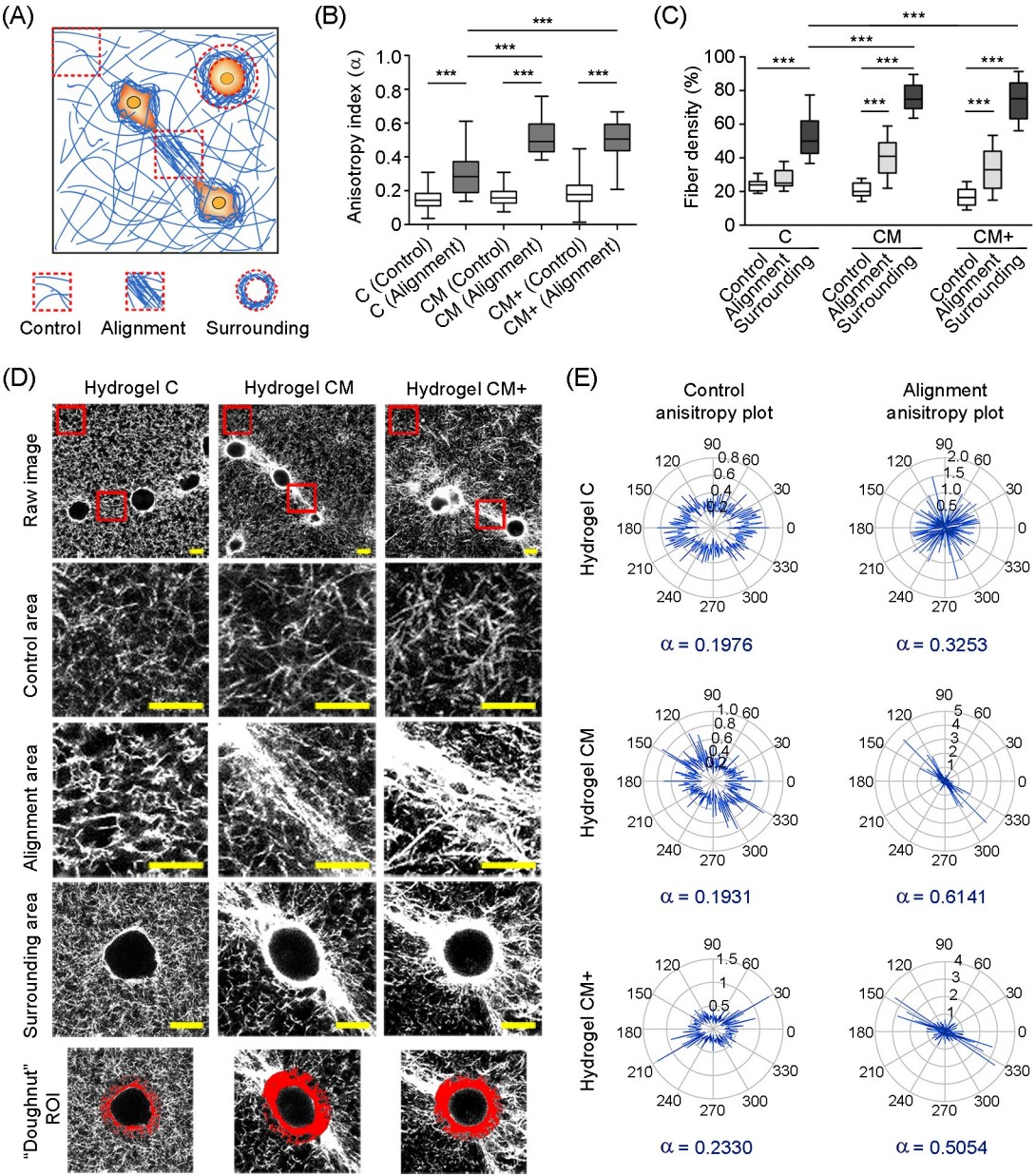

**Fig 6. Quantification of hydrogel remodeling.** (A) Schematic diagram showing cells (orange), collagen fibers (blue), and the location of theControl, Alignment and Surrounding areas used in the quantification. (B) Control and alignment anisotropy ($\alpha$) measured in each hydrogel type. (Number of images, n = 21). (C) Densification results in control, alignment and surrounding areas of cells. (Number of images, n = 21). *** Indicates highly statistically significant difference between groups (p<0.001). (D) Representative images of hydrogels C, CM, and CM+. Scale bar = 10 microns. (E) Polar plots and anisotropy coefficients of the representative images shown in D.

mature FAs as shown in magnified boxes (**Fig 8A**). The trend measured in cells laying on top of the hydrogels was also quantitatively measured in H1299 cells migrating within 3D hydrogels (**Fig 8D and 8E**), where the number and the volume of FA clusters was higher in cells migrating within hydrogels containing Matrigel than in those migrating in collagen-only matrices (p<0.01). Namely, vinculin clusters were barely visible at the protrusions of cells embedded in collagen-only, C hydrogels (**Fig 8F**), whereas, in mixed matrices the presence of

Table 6. Quantification of hydrogel fiber alignment.

| Hydrogel Type | Control Anisotropy | Alignment Anisotropy |
|---|---|---|
| C | 0.151 (0.015) | 0.298 (0.030) |
| CM | 0.167 (0.013) | 0.509 (0.023) |
| CM+ | 0.198 (0.023) | 0.493 (0.026) |

Anisotropy index ($\alpha$) in Control and Alignment areas. Mean and standard error (SEM) of the anisotropy index after 24 hours of 20% FBS stimulated migration in hydrogels C, CM, and CM+. (Number of images, n = 21).

Matrigel promoted the expression of vinculin at FA sites. This suggests a similar role of focal adhesions in 2D and 3D environments related to hydrogel composition and matrix stiffness.

## H1299 cells express higher levels of β1 and β3 in Matrigel-containing hydrogels compared to collagen only hydrogels

Our measurement of the expression level of integrins β1 and β3 in H1299 cells retrieved from hydrogels C, CM, and CM+ were obtained by flow cytometry (**Fig 9**). Our ANOVA test determined a significant interaction between integrin and hydrogel type ($p < 0.004$) on the expression level and also significant main effects of protein ($p < 0.002$) and of hydrogel type ($p < 0.002$). Furthermore, Bonferroni corrected post-hoc analyses revealed that the expression level was significantly higher for integrin β1 than for β3 and also that the expression of both integrins was also significantly higher for hydrogels CM and CM+ than for hydrogel C. In the case of β1, the expression level for hydrogel CM was also significantly higher than for CM+. H1299 cells cultured in flasks were also analyzed by flow cytometry in order to determine the basal expression of integrins and were further compared by means of a U-Mann-Whitney test. Results are shown in **Fig 9A–9C**. We found that integrin β1 and β3 surface expression was not increased in hydrogels of type C ($p > 0.05$) compared to control cells cultured in flasks. Contrarily, integrin β1 expression increased in cells embedded in hydrogels CM and CM+ ($p < 0.05$) compared to control cells grown on flasks, and also compared to cells embedded in hydrogel C ($p < 0.05$). Finally, β3 integrin expression seemed to be higher in cells embedded in hydrogels CM and CM+ compared to cells grown on flasks although the differences were not statistically significant ($p > 0.05$).

The ability of H1299 cells to adhere to the major components of the hydrogels, collagen I and IV and fibronectin, is shown in **S1 Fig**. Notably, the percentage of cell adhesion was higher in the case of collagen I and collagen IV compared with fibronectin ($p < 0.001$). These results are in line with the integrin expression profile observed by flow cytometry (**Fig 9**), where the expression of integrin β1, the main ligand of collagen, was higher compare to β3, the main receptor of fibronectin.

## H1299 cells exert larger forces in hydrogels containing higher concentration of Matrigel

Five cells were imaged using confocal fluorescence microscopy, along with their surrounding fiber mesh, imaged using confocal reflection microscopy, while embedded in each hydrogel type (C, CM, and CM+). Z-stacks where acquired before and after force relaxation using a drug mix based on Nocodazole 1mM, a depolymerizing drug of the tubulin cytoskeleton, and Blebbistatin 50μM, which inhibits the actomyosin contractility. Then the displacement and force fields caused by cell relaxation were calculated using the SAENO software, along with the contractility of the cell, estimated as the total projected force pointing toward the cell center.

**Table 7. Quantification of hydrogel fiber densification.**

| Hydrogel Type | Control Area | Alignment Area | Surrounding Area |
|---|---|---|---|
| C | 0.239 (0.008) | 0.300 (0.020) | 0.518 (0.019) |
| CM | 0.206 (0.009) | 0.369 (0.020) | 0.757 (0.013) |
| CM+ | 0.170 (0.011) | 0.300 (0.027) | 0.746 (0.020) |

Mean and standard error (SEM) of the collagen density quantified after 24 hours of 20% FBS stimulation in hydrogels C, CM, and CM+. (Number of images, n = 21).

Representative images containing cells surrounded by a fiber collagen mesh before (Normal) and after cell relaxation (Relaxed) are shown in **Fig 10B**. Representative examples of the displacement field of the collagen network and the corresponding traction force reconstructions, for the three hydrogel types, are displayed in **Fig 10C**. The average contractility is shown in **Table 8** and **Fig 10A**. Summarizing our results, cell traction forces and contractility increased significantly with Matrigel concentration in our hydrogels. These cell traction forces point towards the center of the cells, following the direction of the retraction of the protrusions.

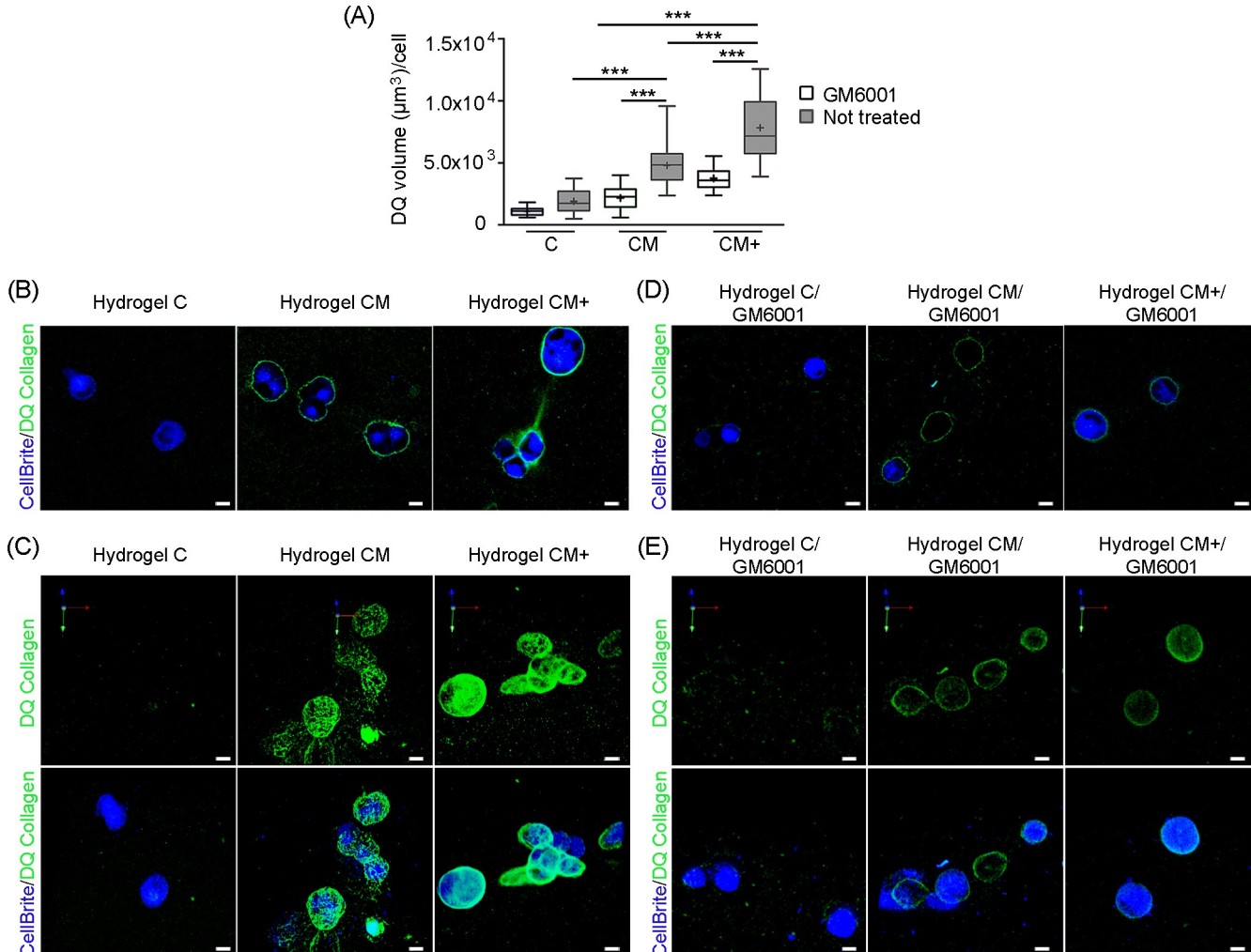

**Fig 7. Quantification of hydrogel degradation.** (A) Quantification of DQ-collagen degradation in hydrogels C, CM, and CM+ with and without GM6001 treatment. (Number of images, n = 30). *** Indicates highly statistically significant difference between groups (p<0.001). (B) 2D and (C) 3D representative images showing DQ-collagen degradation and CellBrite labeled H1299 cells (no GM6001 treatment). (D) 2D and (E) 3D representative images showing DQ-collagen degradation and CellBrite labeled H1299 cells under GM6001 treatment. Scale bar = 10 microns.

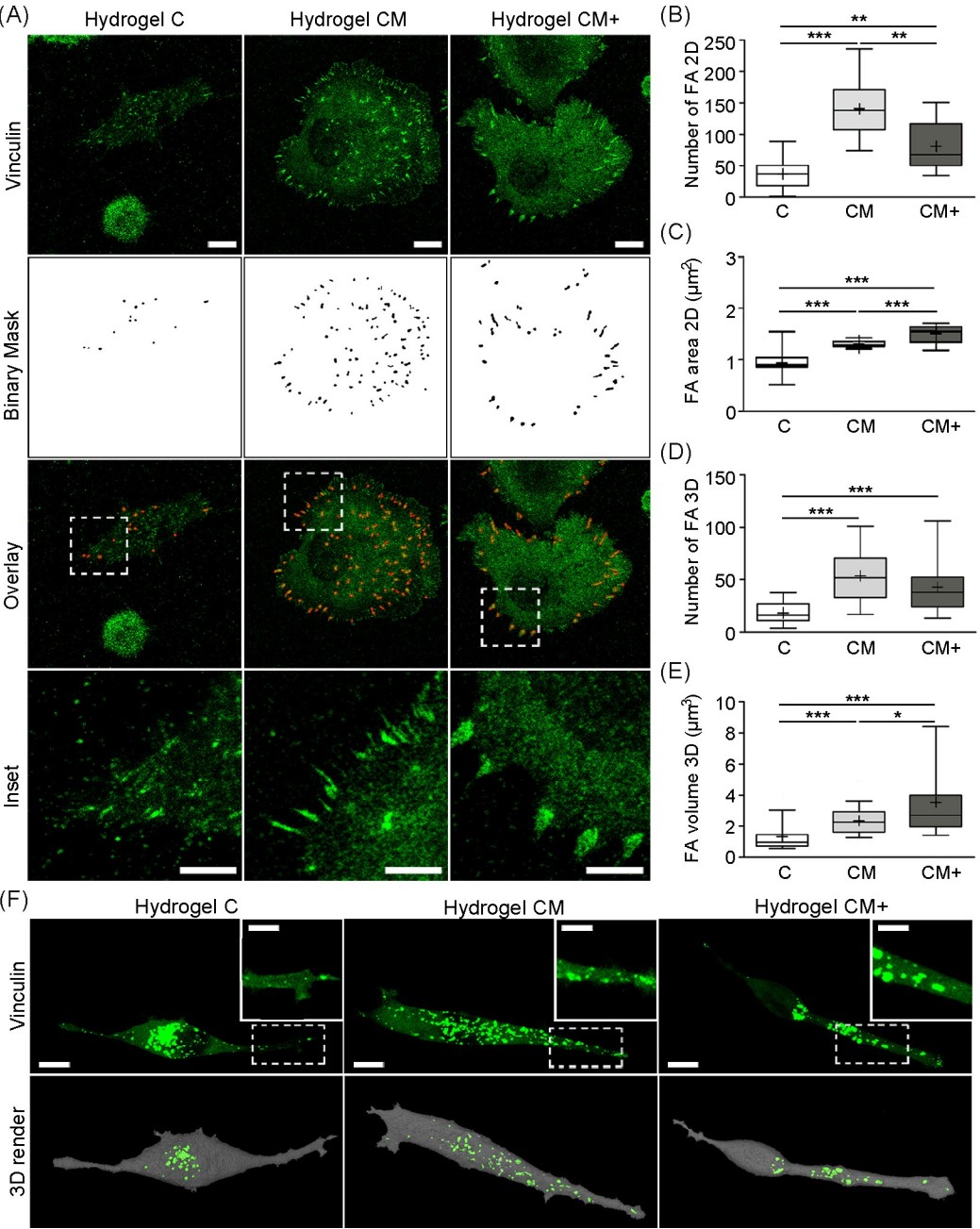

**Fig 8. Quantification of focal adhesions.** (A) Representative images of vinculin stained H1299 cells seeded on top of 2D hydrogels. Scale bar = 10 microns. Binary masks show automatically segmented vinculin patches. White insets show representative, zoomed-in views of FAs. (Scale bar = 5 microns). (B) Number of focal adhesions per cell, quantified on cells seeded on 2D substrates (Number of images, n = 20). (C) Mean area of focal adhesions (Number of images, n = 20). (D) Number of FAs per cell, quantified in H1299 cells embedded in 3D hydrogels (Number of images, n = 25). (E) Average volume of FAs measured in 3D environments (Number of images, n = 25). (F) Representative maximum intensity projection images of 3D H1299 cells stably transfected with mCherry-vinculin plasmid embedded in hydrogels (C, CM, CM+). Scale bar = 10 microns. White insets show representative FA magnified areas at cell protrusions. (Scale bar = 5 microns). 3D renders show vinculin clusters at the protrusion structures resulting from FA segmentation using Fiji and rendering with Amira. *** Indicates highly statistically significant difference between groups (p<0.001). ** Indicates statistically significant difference between groups (p<0.01). * Indicates statistically significant difference between groups (p<0.05).

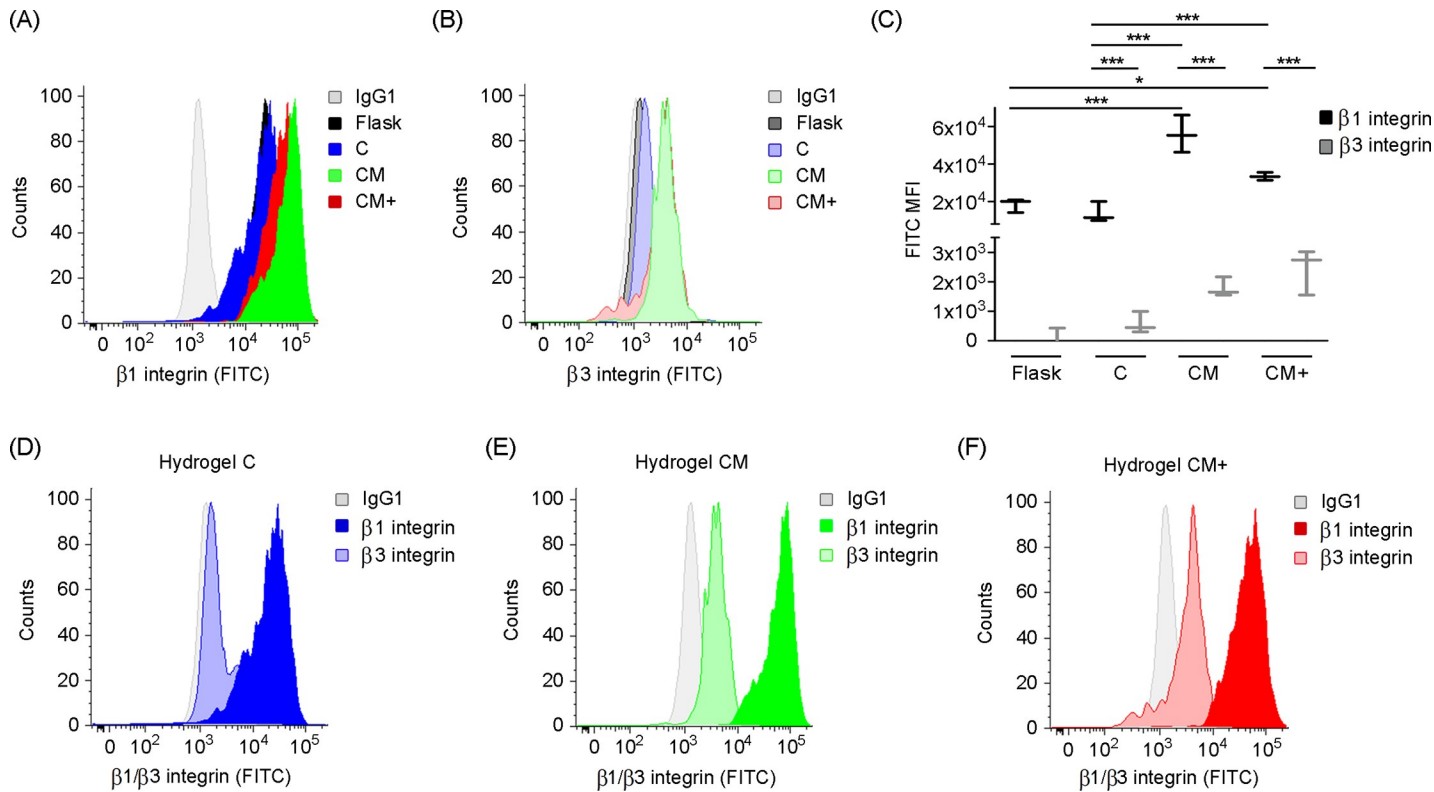

**Fig 9. Flow cytometry-based quantification of integrin expression.** (A) H1299 β3 surface integrin expression in cell culture flask and embedded in hydrogels C, CM, and CM+. (B) H1299 β1 surface integrin expression in cell culture flask and embedded in hydrogels C, CM, and CM+. (C) H1299 β1 and β3 surface integrin expression comparison in cell culture flask and embedded in hydrogels C, CM, and CM+. Number of experiments, n = 3. * Indicates statistically significant difference between groups (p<0.05). (D), (E), (F) H1299 surface β1 and β3 integrin expression compared per hydrogel type (C, CM, and CM+, respectively).

## H1299 Cells in Matrigel-containing hydrogels display thicker, dynamic filopodia

**Fig 11A** shows representative examples of maximum intensity projections (MIPs) of serum stimulated LifeAct-GFP transfected H1299 cells embedded in the three hydrogels used in this study. As clearly seen from the images, H1299 cells in collagen only, C hydrogels display relatively long, thin filopodia. Cells embedded in hydrogels of type CM display few but prominent -i.e. thick, and long- filopodia. Cells embedded in hydrogels of type CM+ display a mixed phenotype, combining thin protrusions similar to those seen in C hydrogels but of shorter length, with thicker, prominent protrusions as those displayed in CM hydrogels. These filopodia-like protrusions were confirmed as filopodia by the co-localization of the actin-bundling protein fascin, with these actin enriched structures (**Fig 11B**). The analysis of the dynamics of the filopodia (**S1**–**S9** Videos) show that the long, thin filopodia seen in C hydrogels (**S1**–**S3** Videos) are almost static during the duration of the experiment (1 hour), while most filopodia in CM (**S4**–**S6** Videos) and especially in CM+ (**S7**–**S9** Videos) hydrogels show a fast pattern of filopodial elongation and retraction.

## Discussion

We have presented a comprehensive analysis of H1299 lung cancer cell to ECM interactions within mixed collagen-Matrigel hydrogels that mimic different microenvironments, ranging from normal connective tissue (C hydrogels) to the leading edge of tumor invasion, at the

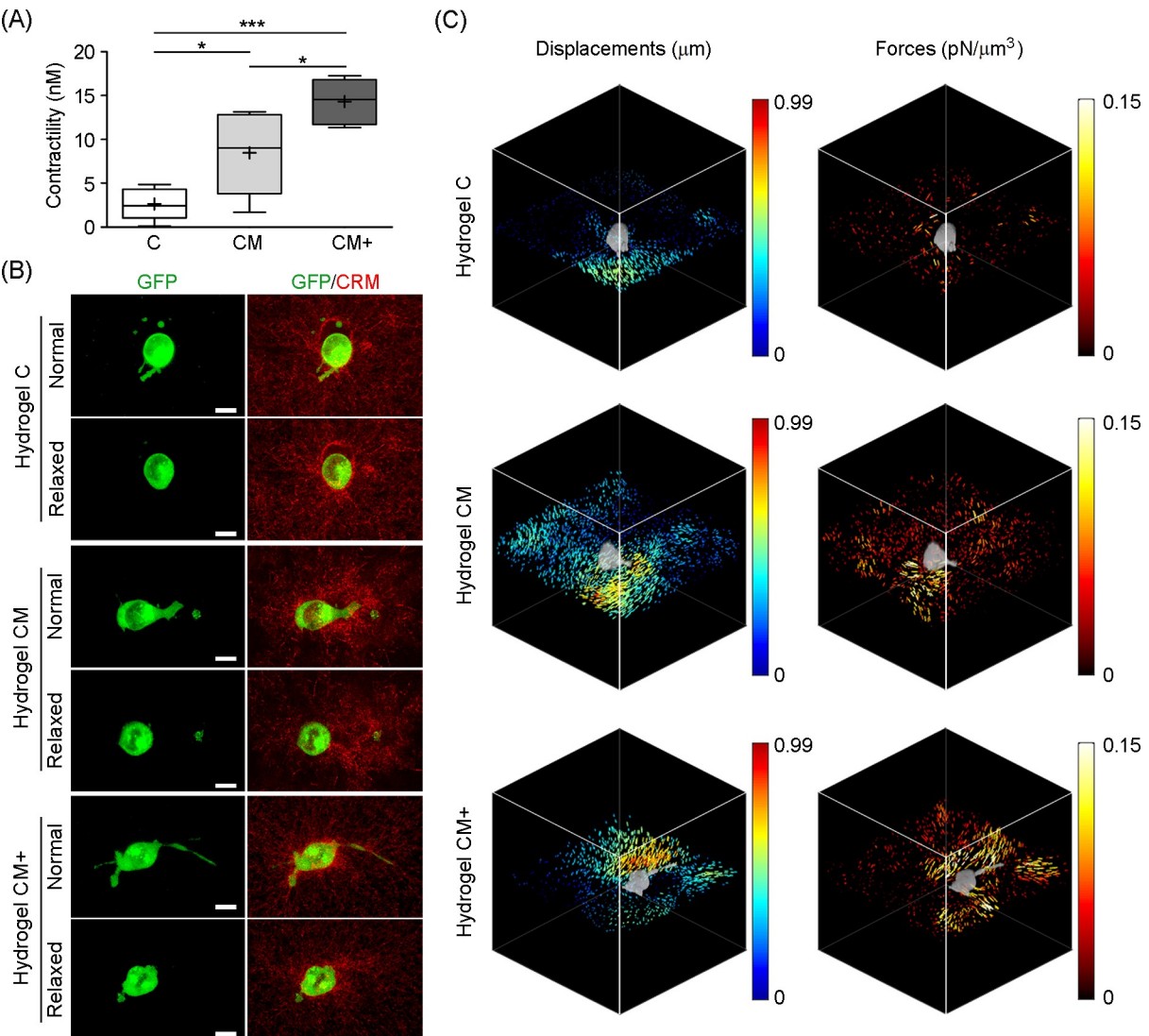

**Fig 10. Quantification of traction forces exerted by H1299 cells embedded in 3D hydrogels.** (A) Average contractility (nN) measured within the three hydrogels types (C, CM, and CM+). *** Indicates highly statistically significant difference between groups (p<0.001). * Indicates statistically significant difference between groups (p<0.05). (Number of experiments, n = 5). (B) Representative confocal images of H1299 cells (green) embedded in each hydrogel and the fiber mesh (red) before (Normal) and after (Relaxed) cell relaxation using 1mM Nocodazole and 50µM Blebbistatin. (C) 3D plots corresponding to the cells shown in b indicating the total displacements and the cell exerted forces in the three hydrogel types (C, CM, and CM+). Arrow hue indicates the magnitude of the displacements. Volume renderings of the representative cells (grey) are overlaid on top of the plots in order to visualize cell-exerted displacements and forces. Images were obtained with the SAENO [27] visualization software in MATLAB.

**Table 8. Traction force microscopy-based quantification of cell contractility.**

| Measurement | Hydrogel C | Hydrogel CM | Hydrogel CM+ |
|---|---|---|---|
| **Mean contractility (nN)** | 2.622 | 8.453 | 14.31 |
| **SEM** | 0.81 | 2.12 | 1.15 |

Cell contractility (nN) measured in 5 H1299 cells within hydrogels C, CM, and CM+ using SAENO software.

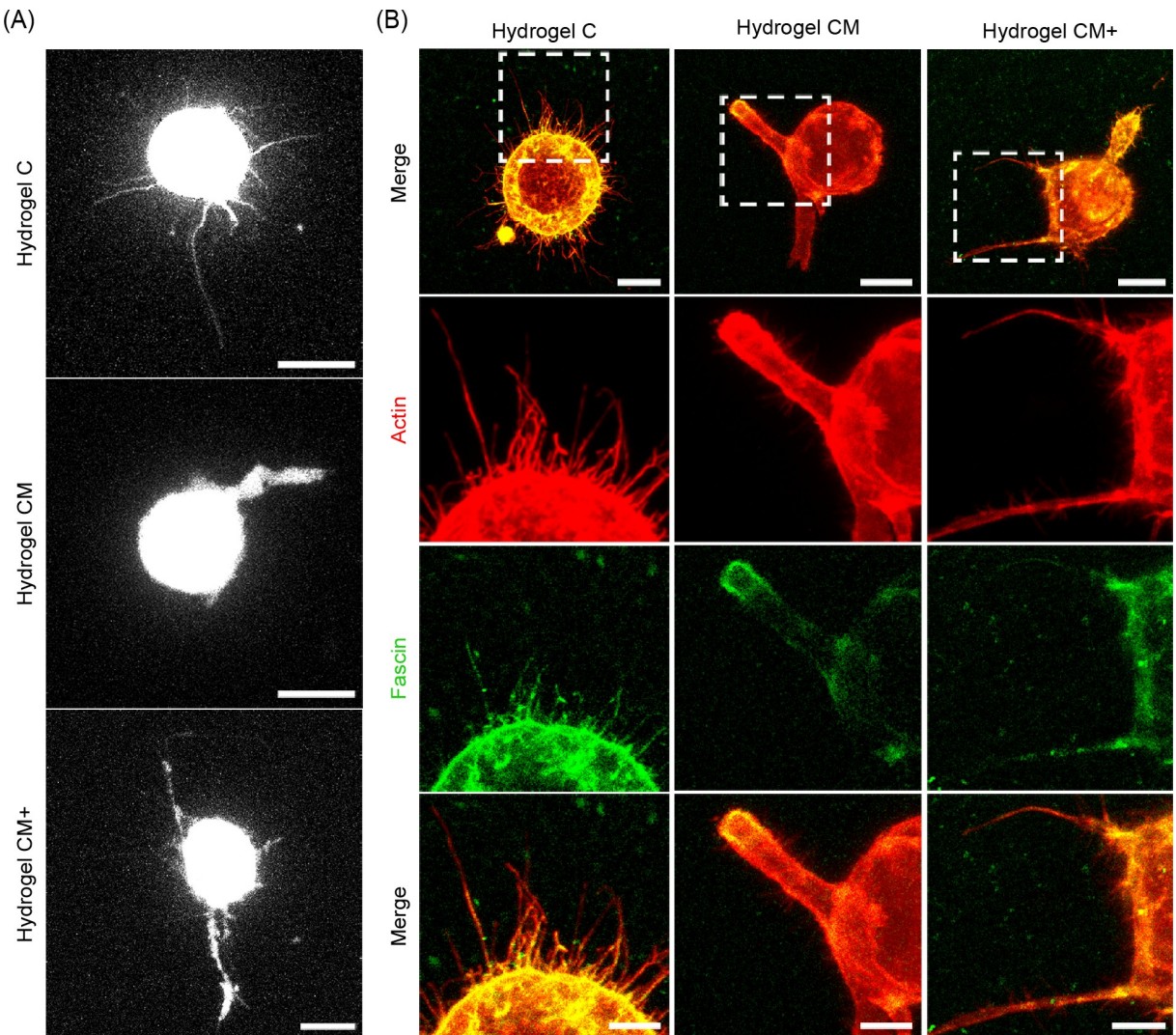

**Fig 11. Filopodial morphology.** H1299 cells displayed different filopodial morphology within the three hydrogels types. (A) Representative maximum intensity projections of H1299-LifeAct transfected cells embedded in the three hydrogel types (C, CM, CM+) after Nocodazole removal and 20% serum stimulation (1 hour). Scale bar = 10 microns. (B) Representative images of H1299 cells embedded within the three hydrogel types stained with phalloidin (red) and fascin (green) after serum stimulation. Scale bars = 10 microns. White insets shown magnified areas with strong F-actin and fascin co-localization at protrusion sites. Scale bars = 5 microns.

boundary between the connective tissue and an increasingly disorganized basement membrane (CM and CM+ hydrogels) [9–11]. This study provides a meaningful explanation of our previous observation that collagen mixed with Matrigel, at an intermediate concentration (CM), favors cell migration compared to collagen only (C) or collagen mixed with Matrigel at high concentration (CM+) [11]. To properly interpret our results, we first characterized the morphology and mechanical properties of our hydrogels. Our measurements confirmed the increased pore size and heterogeneity of Matrigel-containing hydrogels. Compared to our previous work [11], the use of confocal scanning microscopy imaging of TAMRA labelled collagen provides a more accurate quantification of the pore size and fiber length compared to the one obtained using confocal reflection microscopy on unstained hydrogels. The mechanical characterization of the hydrogels, performed using oscillatory stress sweep assays, confirmed

the increased rigidity caused by Matrigel, being significantly higher in hydrogels with the highest concentration of Matrigel (CM+). Moreover, we show for the first time a nonlinear -strain stiffening- behavior in Matrigel-containing hydrogels that had been previously shown in collagen only, C hydrogels [28]. This behavior is significantly more pronounced in hydrogels with high concentration of Matrigel (CM+) than in hydrogels with intermediate concentration (CM) or no Matrigel (C). In summary Matrigel causes larger pore sizes and heterogeneity on the underlying collagen mesh and increases the rigidity of the hydrogel while enhancing a nonlinear strain stiffening effect.

Our quantification of DQ-collagen I cleavage caused by H1299 proteolytic activity during cell migration showed increased matrix degradation with increasing Matrigel concentration. This can be explained by the higher matrix stiffness, which is known to trigger MMP activity in most tumor cell types [29], although it cannot be discarded that the presence of soluble growth factors found in Matrigel may also stimulate MMP activity [30]. Consequently, DQ-collagen I degradation, in the presence of the GM6001 metalloproteinase inhibitor, decreased sharply in Matrigel-containing hydrogels, compared to collagen I-only hydrogels, and MMP-blockade using GM6001 produced a significant reduction of H1299 cell migration speed, as well as a reduction in H1299 cell invasive capabilities in Matrigel-containing hydrogels, not seen in collagen only hydrogels. Interestingly, the increased migration rates observed in Matrigel-containing hydrogels, could be explained by the generation of tunnel-like paths that facilitate collective cell migration through these hydrogels. In line with these results, Wolf *et al*. reported how malignant cells migrating through 3D matrices can proteolytically remodel the ECM and generate microtracks that trigger collective cell invasion [31]. Summarizing, H1299 cells respond to the increased rigidity of Matrigel-containing hydrogels by enhancing MMP-based degradation of the surrounding matrix.

Integrin expression at the cell surface is related to the ability of the cells to attach to and remodel the surrounding ECM, which in turn affects their migration capacity. Indeed, it has been shown that ECM stiffness triggers the expression of integrins and the assembly of FA at the cell surface [32]. In our flow cytometry assays, β1 integrin expression was found at higher levels in H1299 cells previously embedded in hydrogels CM and CM+ compared to cells embedded in hydrogel C. The fact that the expression of β1, a known ligand for collagen I and collagen IV, increases with increasing stiffness is in accordance with previous studies that demonstrate that soft substrates induce integrin internalization and subsequent proteasomal degradation [32,33]. No significant differences in the level of integrin β3 expression were found among the cells cultured inside the three types of hydrogels. These results indicate that H1299 integrin expression profile is modulated by hydrogel composition, being Matrigel-containing hydrogels (CM and CM+) the ones displaying the highest levels of β1 integrin expression. In this sense, our adhesion to ECM protein substrates experiments confirmed that H1299 cells adhere primarily to collagen I and collagen IV, but not significantly to fibronectin. These results agree with the enhanced β1 integrin expression found in Matrigel-containing hydrogels (CM and CM+), since collagen I and collagen IV are important components of these hydrogels. This supports the idea that Matrigel containing hydrogels suffer increased ECM degradation caused by their high levels of β1 integrin expression In agreement with this, several groups have reported in other cell types that β1 integrin co-clusters at the cell membrane with several MMPs and drives ECM degradation [34,35].

The increased cancer cell integrin expression found in Matrigel-containing hydrogels nicely correlates with our analysis of focal adhesions in H1299 cells, which showed higher number of FAs in cells cultured on Matrigel containing hydrogels. This agrees with previous studies that show how, in tumor cells, vinculin is required for FA formation and regulates the clustering of integrin at cell membrane [36]. Furthermore, the area of the FAs increased with increasing

Matrigel content. Previous migration studies done on 2D substrates have shown that there is a biphasic relationship between the size of FAs and cell speed [37], i.e. cell speed steadily increases until a threshold value in FA size is reached, beyond which cells slow down. This points at a balance between adhesion and migration speed, which indicates that larger, mature FAs lean favor attachment while small, less mature FAs promote cell migration. This is supported by our quantification of migration speed in our hydrogels, where we found a similar relationship between cell speed and the size of FAs. According to this, the size of FAs in our high-concentration CM+ hydrogels may be above the threshold between both migration phases. Supporting this interpretation, Palecek *et al*. reported that substrates with a high concentration of ligands elicit stronger bonds between cells and matrix that hinder cell migration [38,39].

Our quantification of ECM remodeling showed that collagen fiber alignment was higher in Matrigel-containing hydrogels (CM and CM+) compared to collagen only hydrogels (C), indicating that hydrogel composition and stiffness are crucial determinants of the extent of matrix remodeling. Indeed, H1299 cells were able to align the collagen fibers in all three hydrogels, a phenomenon that has been associated with malignancy and cell invasiveness [40]. These results are similar to previous studies reported for different types of cancer including glioblastoma (U87 tumor spheroids) [41] and breast cancer (tumors in vivo) [42]. Similar to what occurs with fiber alignment, the densification of the matrices in the surrounding region of the cells was higher in Matrigel-containing hydrogels. In agreement with the relationship found between the size of FAs and the extent of ECM remodeling, previous studies using fibroblast have demonstrated how "super-mature" FAs are involved in a strong remodeling of ECM, as well as in an efficient fiber contraction by providing a strong adhesion to the mesh [43].

To confirm that both the alignment and densification of the ECM at the cell-matrix boundary are caused by forces exerted by the cells [44], we performed 3D TFM assays to quantify cellular forces exerted by H1299 cells within our hydrogels. The magnitude of the obtained forces was similar to what had been previously reported using TFM [27] and optical methods [45]. Furthermore, we found a linear relationship between H1299 cell contractility and matrix stiffness. This trend agrees with previous studies on cell-exerted forces over 2D substrates reporting that cell tractions increase with higher substrate stiffness [46]. Contradicting these results, Steinwachs *et al*. quantified MDA-MB-231 3D traction forces within collagen I hydrogels of increasing concentration, where the Young Modulus increased from 44 Pa to 513 Pa and the average pore size decreased [27]. They reported similar traction forces across all hydrogels. To save the apparent contradiction with the expected behavior seen in 2D [46], they explained their results by the increasingly smaller pore size found in denser 3D matrices. The argument used was that cells in small pore matrices spread less and generate thinner protrusions than in larger pore sized matrices, thus exerting lower magnitude traction forces. The fact that our Matrigel-containing hydrogels display the opposite relationship between stiffness and pore size seems to support this argument. Indeed, the increased forces exerted by the cells, caused by the increased stiffness of our Matrigel-containing hydrogels, favors migration due to the existence of larger pores.

The stronger cell forces exerted by H1299 cells in our Matrigel-containing hydrogels (CM and CM+) are in accordance with the highest degree of remodeling seen in these hydrogels as well as with the increased expression of β1 integrin and larger FA clusters at the cell membrane. Previous studies have described how vinculin expression at FAs engages the actin cytoskeleton with the ECM at protrusion sites in order to exert tensile forces and contract the cell body during cell migration [36,47]. Thereby, the lower number and size of FAs found in collagen-only C hydrogels yields a reduced cytoskeletal contraction and a speed reduction during cell migration. Likewise, Goffin *et al*. described a linear relationship between the size and the

force exerted by the FAs on 2D substrates [43], where larger, mature FAs yielded higher tensile forces. This evidence agrees with our results obtained within C, CM, and CM+ hydrogels, in which cell contractility increases as the size of the vinculin clusters increases. In our view, the fact that H1299 cells are able to move much faster in CM hydrogels than in CM+ hydrogels relates back to the biphasic relationship between the size of FAs and speed seen in 2D [37]. This indicates that large, mature FAs more effectively promote cell attachment forces while tractions exerted through small, less mature FAs promote cell migration. This is also supported by studies [48] that show that cell migration within 3D environments depends on an optimal balance between cell adhesion and contraction. Accordingly, in our CM+ hydrogels the cells seem to be exerting strong forces that are more functionally related to attachment than to cell traction.

Finally, the qualitative analysis of the filopodia morphology and dynamics of H1299 cells seems also consistent with our previous observations. In collagen only C hydrogels, the cells' larger, thin and mostly static filopodia seem just appropriate given the lower number of FAs and reduced cell-ECM remodeling activity. In CM and CM+ hydrogels however, thick filopodia with faster dynamic properties appear more appropriate for the increased degradation seen in those hydrogels and to promote effective cell migration. Supporting these results, Liou *et al.* reported a similar behavior in lung cancer cells where soft substrates elicited long and dense filopodia, used to explore the surrounding environment, whereas stiffer ones promoted shorter protrusions that seem to be involved in tumor transformation [49].

Bearing all our findings in mind, H1299 lung cancer cell migration inside CM hydrogels that recapitulate the microenvironment at the leading edge of tumor invasion is favored by increased integrin expression and metalloproteinase activity, which in turn is mainly stimulated by the architecture and the mechanical properties of the hydrogels. It should be kept in mind that the presence of soluble growth factors found in Matrigel may have an influence on different cellular processes studied in this work, such as cell adhesion to ECM or cell migration. However, the concentration of these factors in our hydrogels is far from the experimental doses reported in the literature, as well as with the effective dose ($ED_{50}$) recommended by research companies (**S2 Table**).

Our results quantitatively show the role of hydrogel composition in promoting cell migration, since, in pure collagen matrices (C hydrogels), cell migration decreases, possibly due to reduced pore size as well as due to a fewer number of focal adhesions and weaker traction forces to sustain an effective migration. Contrarily, cell motility in hydrogels containing the highest Matrigel concentration (CM+ hydrogels) is hindered by an excessive attachment which is not sufficiently compensated by the increased remodeling and forces exerted. These results nicely relate to our previous cancer cell migration results [11], providing important information about the role of the ECM properties, the remodeling capacity of the cells, including the quantification of cell-exerted forces.

## Supporting information

**S1 Table. Main SAENO software parameter used for TFM acquisition.**
(DOCX)

**S2 Table. Main growth factors presented in Matrigel and its concentrations.**
(DOCX)

**S1 Fig. Adhesion assay to ECM protein substrates.** H1299 cells were seeded over collagen I, collagen IV and fibronectin coatings. BSA was used as negative adhesion control. *** Indicates highly statistically significant difference between groups (p<0.001) compared by one-way

ANOVA analysis. Four wells were analyzed per ECM protein and experiment. The experiment was performed in triplicate.
(TIF)

**S1 Video. Representative video of a sequences of maximum intensity projection of a Z-stack containing a H1299-LifeAct cell within hydrogel type C.**
(AVI)

**S2 Video. Representative video of a sequences of maximum intensity projection of a Z-stack containing a H1299-LifeAct cell within hydrogel type C.**
(AVI)

**S3 Video. Representative video of a sequences of maximum intensity projection of a Z-stack containing a H1299-LifeAct cell within hydrogel type C.**
(AVI)

**S4 Video. Representative video of a sequences of maximum intensity projection of a Z-stack containing a H1299-LifeAct cell within hydrogel type CM.**
(AVI)

**S5 Video. Representative video of a sequences of maximum intensity projection of a Z-stack containing a H1299-LifeAct cell within hydrogel type CM.**
(AVI)

**S6 Video. Representative video of a sequences of maximum intensity projection of a Z-stack containing a H1299-LifeAct cell within hydrogel type CM.**
(AVI)

**S7 Video. Representative video of a sequences of maximum intensity projection of a Z-stack containing a H1299-LifeAct cell within hydrogel type CM+.**
(AVI)

**S8 Video. Representative video of a sequences of maximum intensity projection of a Z-stack containing a H1299-LifeAct cell within hydrogel type CM+.**
(AVI)

**S9 Video. Representative video of a sequences of maximum intensity projection of a Z-stack containing a H1299-LifeAct cell within hydrogel type CM+.**
(AVI)

## Author Contributions

**Conceptualization:** Carlos Ortiz-de-Solorzano.

**Data curation:** María Anguiano, Xabier Morales.

**Formal analysis:** María Anguiano, Xabier Morales, Martín Martínez.

**Funding acquisition:** Carlos Ortiz-de-Solorzano.

**Investigation:** María Anguiano, Xabier Morales, Carlos Castilla, Alejandro Rodríguez Pena, Cristina Ederra, Maider Esparza, Hippolyte Amaveda, Mario Mora, Nieves Movilla.

**Methodology:** María Anguiano, José Manuel García Aznar, Iván Cortés-Domínguez.

**Project administration:** Carlos Ortiz-de-Solorzano.

**Software:** María Anguiano, Mikel Ariz.

**Supervision:** Carlos Ortiz-de-Solorzano.

**Validation:** María Anguiano.

**Visualization:** María Anguiano, Xabier Morales.

**Writing – original draft:** María Anguiano, Carlos Ortiz-de-Solorzano.

**Writing – review & editing:** José Manuel García Aznar, Iván Cortés-Domínguez, Carlos Ortiz-de-Solorzano.

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
