## [Decision Letter · Decision Letter 0]

24 Sep 2019

PONE-D-19-18819

The use of mixed collagen-Matrigel matrices of increasing complexity recapitulates the biphasic role of cell adhesion in cancer cell migration: ECM sensing, remodeling and tractions at the leading edge of cancer invasion

PLOS ONE

Dear Dr. Ortiz-de-Solorzano,

Thank you for submitting your manuscript to PLOS ONE. After careful consideration, we feel that it has merit but does not fully meet PLOS ONE’s publication criteria as it currently stands. Therefore, we invite you to submit a revised version of the manuscript that addresses the points raised during the review process.

As you will see, both reviewers agreed that your manuscript is of interest but they also raised a number of important remarks that should be addressed to move your manuscript foward.

We would appreciate receiving your revised manuscript by Nov 08 2019 11:59PM. To enhance the reproducibility of your results, we recommend that if applicable you deposit your laboratory protocols in protocols.io, where a protocol can be assigned its own identifier (DOI) such that it can be cited independently in the future. For instructions see: http://journals.plos.org/plosone/s/submission-guidelines#loc-laboratory-protocols

We look forward to receiving your revised manuscript.

Kind regards,

Daniel Bouvard, Ph.D.

Academic Editor

PLOS ONE

Journal Requirements:

1. Please amend your list of authors on the manuscript to ensure that each author is linked to an affiliation. Authors’ affiliations should reflect the institution where the work was done (if authors moved subsequently, you can also list the new affiliation stating “current affiliation:….” as necessary).

Reviewers' comments:

Reviewer's Responses to Questions

**Comments to the Author**

1. Is the manuscript technically sound, and do the data support the conclusions?

Reviewer #1: Partly

Reviewer #2: Partly

2. Has the statistical analysis been performed appropriately and rigorously? 

Reviewer #1: Yes

Reviewer #2: Yes

3. Have the authors made all data underlying the findings in their manuscript fully available?

Reviewer #1: No

Reviewer #2: Yes

4. Is the manuscript presented in an intelligible fashion and written in standard English?

Reviewer #1: Yes

Reviewer #2: Yes

5. Review Comments to the Author

Reviewer #1: This manuscript desribes an interesting and relevant (somewhat) bi-phasic relationship between ECM composition, rigidity, and cell characteristics (adhesion morphology, migration/invasion). In general, the work is interesting, although fairly similar studies have been published with similar results.

There are some concerns with some of the experimental approaches and more impactful concerns about how the data are presented. These are detailed below.

- The use of microtubule disruption via nocodazole to attain a zero-force state seems, at best, circuitous and, at worst, erroneous. True that peripheral projections will retract, but countless studies have shown that such disruption increases cell contractility through loss of outward/peripheral counter-tension to centrpetal actomyosin contractility. Thus, the use of this approach should be better validated (through either literature or supplemental data) or corroborated using an approach that more directly eliminates tension (e.g. blebbistatin or an inhibitor of Rho kinase).

- The primary data exemplifying a difference in adhesion size in 2D culture are not particularly convincing. The data are what they are, and the quantification suggests relevance, but the micrographs are simply not convincing. Perhaps magnified insets or better examples can be used.

- More importantly, the data on adhesion size in 2D doesn't seem particularly relevant to the works central thesis of the effects of ECM mechanics on cell behavior in 3D. In keeping with this and the aforementioned concern, the figures depicting 3D adhesion structure are practically uninterpretable - they appear blown out/over-exposed and offer no basis for real comparison. The authors are encouraged to look in the considerable literature available (for exmaple from Ken Yamada's or Chris Chen's groups) for examples of how to depict 3D adhesions.

- More of the primary data should be included (more examples of adhesion morphology; 3D renderings of adhesions and tractions in 3D) as supplements.

- The TFM data should be scaled globally (i.e. all three examples should be shown using the same TFM value scale) to better accentuate the relevant differences.

- The authors claim a difference in filopodia formation, but the only method used to visualize projections is LifeAct. Also - or perhaps therefore - many of the 'filopodia/ appear more like robust projections than true filopodia. Either the wording/intrepretation should be substantially edited or the authors should confirm their findings using a better filopodial marker (e.g. MyoX).

Reviewer #2: The manuscript from Anguiano describing H1299 lung cancer cells behaviour in various collagen/matrigel matrices is interesting and valuable despite its mainly technological and descriptive approach. Experiments are generally rigorously performed and analysed. The main point addressed by the authors is the impact of matrix stiffness on cell behaviour, however, the use Matrigel raises important questions because it introduces a bias linked to its bio-active composition (growth factors and ECM proteins). This could have influenced any of the parameters analysed in the study (migration, MMPs, integrin expression, FAs, filopodia..) except those concerning stiffness measurements of the acellularized scaffolds. The model is interesting but the parameters affecting the cells must be considered. Authors mention “small sized focal adhesions and larger focal adhesions” in their abstract but such adhesion structures are not shown, neither quantified. Filopodia do not belong to focal adhesions.

Authors should take these comments in consideration all the way through their manuscript abstract and should adapt their interpretations accordingly.

6. PLOS authors have the option to publish the peer review history of their article (what does this mean?). If published, this will include your full peer review and any attached files.

Reviewer #1: No

Reviewer #2: No

---

## [Author Response · Author response to Decision Letter 0]

7 Dec 2019

To all reviewers and the associate editor:

We would like to thank the reviewers for their constructive criticism, and the associate editor for the opportunity to submit this revised version of our manuscript. We have carefully read and addressed, to the best of our capacity, all the reviewers’ comments and we are confident that in this process the quality of the manuscript has greatly improved.

In the following lines, we address the specific concerns of both reviewers:

Response to Reviewer #1

This manuscript describes an interesting and relevant (somewhat) bi-phasic relationship between ECM composition, rigidity, and cell characteristics (adhesion morphology, migration/invasion). In general, the work is interesting, although fairly similar studies have been published with similar results.

We thank the reviewer for his/her positive evaluation of the scientific merit of our work. Regarding the comment that similar studies have been published, we would like to stress the fact that indeed, some studies -most of them referred to in our manuscript- have been published that show similar results. However, those results only look at one or a few aspects of the problem, focus on non-cancer cells, mostly fibroblasts, or do it in non-physiologically relevant setups, while we believe that we have provided a more robust, integrated, comprehensive quantitative analysis of the principal elements involved in the mechanobiology of cancer cell migration (ECM remodelling, integrin expression, proteolytic activity, presence of focal adhesions and traction forces), within a highly physiological environment, properly characterized both morphologically and mechanically.

There are some concerns with some of the experimental approaches and more impactful concerns about how the data are presented. These are detailed below.

- The use of microtubule disruption via nocodazole to attain a zero-force state seems, at best, circuitous and, at worst, erroneous. True that peripheral projections will retract, but countless studies have shown that such disruption increases cell contractility through loss of outward/peripheral counter-tension to centrpetal actomyosin contractility. Thus, the use of this approach should be better validated (through either literature or supplemental data) or corroborated using an approach that more directly eliminates tension (e.g. blebbistatin or an inhibitor of Rho kinase).

We truly appreciate this comment. Indeed, we were not aware of the fact pointed out by the reviewer. Even if some studies seem to indicate that the referred centripetal actomyosin contractility should be low for soft hydrogels as the ones we use [1], we have repeated the TFM experiments using a combination of Nocodazole, to disrupt microtubules and inhibit microtubule dynamics, causing cell protrusions to retract, and Blebbistatin, to impair actin cytoskeletal tension, causing cell relaxation. This drug mix has been used in the literature as it ensures total relaxation of the forces exerted by the cell. This is now described in the Methods section, under “Traction Force Microscopy assays”. More importantly, as shown in the updated Results section, under subsection “H1299 cells exert larger forces in hydrogels containing higher concentration of Matrigel”, the new contractility values follow a more expected trend, as they increase with the rigidity of the hydrogel, to a point where statistically significant differences are found between the forces exerted by the cells in all three hydrogels. 

1. Tabdanov ED, Puram V, Zhovmer A, Provenzano PP. Microtubule-Actomyosin Mechanical Cooperation during Contact Guidance Sensing. Cell Rep. 2018. doi:10.1016/j.celrep.2018.09.030

- The primary data exemplifying a difference in adhesion size in 2D culture are not particularly convincing. The data are what they are, and the quantification suggests relevance, but the micrographs are simply not convincing. Perhaps magnified insets or better examples can be used.

Following the reviewer’s advice, Figure 8 has been recreated. First, better, clearer examples of the three phenotypes have been selected (Figure 8a, “Vinculin” row). Then to more clearly show the differences between them, we now include the segmentation masks that were used in the quantification (Figure 8a, “Binary Mask” row), an overlay of the original images and the segmentation masks (Figure 8a, “Overlay” row), and a zoomed-in view of a relevant, adhesion containing parts of the cells (Figure 8a, “Inset” row). We believe that now, the measured differences in the number and size of the focal adhesions (Figure 8b,c) are more convincingly supported by the figure. 

- More importantly, the data on adhesion size in 2D doesn't seem particularly relevant to the works central thesis of the effects of ECM mechanics on cell behavior in 3D. In keeping with this and the aforementioned concern, the figures depicting 3D adhesion structure are practically uninterpretable - they appear blown out/over-exposed and offer no basis for real comparison. The authors are encouraged to look in the considerable literature available (for exmaple from Ken Yamada's or Chris Chen's groups) for examples of how to depict 3D adhesions.

We agree that the 3D micrographs shown in the original version of the manuscript were far from optimal. Accordingly, we have repeated the experiment and selected new cells that represent better examples of each phenotype. Then, on top of showing 3D renderings (Figure 8f, “3D render” row), as it was done in the original manuscript, we show maximum intensity projections (Figure 8f, “Vinculin” row) of the cells, including magnifying insets located at the protrusion sites, that more clearly show the differences. Furthermore, as it is now described in the Methods section, under “Quantification of vinculin-stained focal adhesions”, we have also quantified the number and volume of adhesions in 3D for our H1229-mCherry-vinculin cells embedded in the 3D hydrogels using a custom-made FIJI plug-in. These new results, that support our 2D observations, are shown in Figure 8d,e, and mentioned in the Results section, under “H1299 cells display a higher number of larger focal adhesions in Matrigel-containing hydrogels compared to collagen only hydrogels”.

- More of the primary data should be included (more examples of adhesion morphology; 3D renderings of adhesions and tractions in 3D) as supplements.

Given the number of experiments done, and the number of images acquired and used to produce our experimental results, including all the original images may saturate the Supporting information section. Instead, we provide (See Supporting information, “Raw and Processed Data”) a link to a Google Drive folder that contains, in an organized, per-experiment way, all the images used, and all the results generated to produce the results shown in our manuscript. The link provided is: https://drive.google.com/drive/folders/1r3nMHaTVVwTh_P5GmRLFYWIkxlY_icOI?usp=sharing. This link is available to the reviewers, and will be left freely available for the readers, if the manuscript is accepted for publication.

- The TFM data should be scaled globally (i.e. all three examples should be shown using the same TFM value scale) to better accentuate the relevant differences.

We thank the reviewer for this comment. Following his/her advice, we now use the same scale for both displacements and forces (See Figure 10c). This way, it is indeed easier to compare and appreciate the differences between phenotypes.

- The authors claim a difference in filopodia formation, but the only method used to visualize projections is LifeAct. Also - or perhaps therefore - many of the 'filopodia/ appear more like robust projections than true filopodia. Either the wording/intrepretation should be substantially edited or the authors should confirm their findings using a better filopodial marker (e.g. MyoX).

Following this reviewer’s advice, we have confirmed the true nature of what we call filopodia by co-staining H1299 cells after filopodia stimulation in all three hydrogels. To that aim, we imaged representative cells labelled with phalloidin, a classical F-actin marker, and the actin-adaptor protein fascin, specifically localized at the filopodia. This is now described in the Methods section, under “High-resolution analysis of filopodial protrusions in migrating H1299 cells”, and the results as shown in the new Figure 11b, which shows co-localization of actin and fascin at the filopodia, thus confirming their true filopodial nature.

Response to Reviewer #2

The manuscript from Anguiano describing H1299 lung cancer cells behaviour in various collagen/matrigel matrices is interesting and valuable despite its mainly technological and descriptive approach. Experiments are generally rigorously performed and analysed. The main point addressed by the authors is the impact of matrix stiffness on cell behaviour, however, the use Matrigel raises important questions because it introduces a bias linked to its bio-active composition (growth factors and ECM proteins). This could have influenced any of the parameters analysed in the study (migration, MMPs, integrin expression, FAs, filopodia..) except those concerning stiffness measurements of the acellularized scaffolds. The model is interesting but the parameters affecting the cells must be considered. Authors mention “small sized focal adhesions and larger focal adhesions” in their abstract but such adhesion structures are not shown, neither quantified. Filopodia do not belong to focal adhesions.

Authors should take these comments in consideration all the way through their manuscript abstract and should adapt their interpretations accordingly.

We thank the reviewer for his/her encouraging words regarding the interest and rigour of our work. Regarding the reviewer’s main point, we agree that, indeed, the complex bio-active composition of Matrigel could affect the results and the interpretation of the obtained mechano-biological results. This was already mentioned in the manuscript, and is now even more relevantly so at the end of the Discussion. Somehow, this accepted level of uncertainty caused by the use of Matrigel is the price that has to be paid in return for being able to perform our experiments in an environment that closely represents the physiology and composition of the microenvironment of migrating cancer cells. However, we believe that the main effect of Matrigel is exerted through the morphological and mechanical differences that it produces in the hydrogels, instead of being caused by the soluble growth factors found in it. This is supported by the fact that the concentration of these factors in Matrigel (thus in our mixed-composition hydrogels) is far from the experimental doses used and reported in the literature, and is also far from the effective dose recommended by the manufacturing companies (See the new supplementary table S2Table). In summary, even if we acknowledge a level of uncertainty caused by the presence of growth factors, we believe that this is minor compared to the effect that Matrigel exerts due to its biomechanical and morphological properties.

Regarding the reviewer’s minor comments, we would like to focus his/her attention to Figure 8, where we do show qualitative and quantitative differences in the size of focal adhesions in our cells, based on vinculin staining, both in 2D and 3D. 

As for the comment on the Filopodia not belonging to focal adhesions. It is correct. Therefore, we have reviewed the text to make sure that we do not include such a statement.

---

## [Decision Letter · Decision Letter 1]

3 Jan 2020

The use of mixed collagen-Matrigel matrices of increasing complexity recapitulates the biphasic role of cell adhesion in cancer cell migration: ECM sensing, remodeling and forces at the leading edge of cancer invasion

PONE-D-19-18819R1

Dear Dr. Ortiz-de-Solorzano,

We are pleased to inform you that your manuscript has been judged scientifically suitable for publication and will be formally accepted for publication once it complies with all outstanding technical requirements.

With kind regards,

Daniel Bouvard, Ph.D.

Academic Editor

PLOS ONE

Additional Editor Comments (optional):

Reviewers' comments:

Reviewer's Responses to Questions

**Comments to the Author**

1. If the authors have adequately addressed your comments raised in a previous round of review and you feel that this manuscript is now acceptable for publication, you may indicate that here to bypass the “Comments to the Author” section, enter your conflict of interest statement in the “Confidential to Editor” section, and submit your "Accept" recommendation.

Reviewer #1: All comments have been addressed

Reviewer #2: All comments have been addressed

2. Is the manuscript technically sound, and do the data support the conclusions?

Reviewer #1: Yes

Reviewer #2: Yes

3. Has the statistical analysis been performed appropriately and rigorously? 

Reviewer #1: Yes

Reviewer #2: Yes

4. Have the authors made all data underlying the findings in their manuscript fully available?

Reviewer #1: Yes

Reviewer #2: Yes

5. Is the manuscript presented in an intelligible fashion and written in standard English?

Reviewer #1: Yes

Reviewer #2: Yes

6. Review Comments to the Author

Reviewer #1: This extensively revised manuscript describes an interesting and relevant bi-phasic relationship between ECM composition, rigidity, and cell characteristics (adhesion morphology, migration/invasion). While similar studies studies exist, there are enough new observations & data here to strengthen & advance the field. In addition, the authors did an excellent job in addressing all of the concerns from the prior round of review. I recommend acceptance of the article for publication.

Reviewer #2: The authors have adequately addressed the questions raised by the reviewer and have made the revision accordingly.

7. PLOS authors have the option to publish the peer review history of their article (what does this mean?). If published, this will include your full peer review and any attached files.

Reviewer #1: Yes: Alan K. Howe

Reviewer #2: No

---

## [Editor Report · Acceptance letter]

10 Jan 2020

PONE-D-19-18819R1 

The use of mixed collagen-Matrigel matrices of increasing complexity recapitulates the biphasic role of cell adhesion in cancer cell migration: ECM sensing, remodeling and forces at the leading edge of cancer invasion. 

Dear Dr. Ortiz-de-Solorzano:

I am pleased to inform you that your manuscript has been deemed suitable for publication in PLOS ONE. Congratulations! Your manuscript is now with our production department. 

With kind regards,

on behalf of

Dr Daniel Bouvard 

Academic Editor

PLOS ONE